# GIQ: Benchmarking 3D Geometric Reasoning of Vision Foundation Models with Simulated and Real Polyhedra

**Mateusz Michalkiewicz**[1], **Anekha Sokhal**[1], **Tadeusz Michalkiewicz**[2], **Piotr Pawlikowski**[2],
**Mahsa Baktashmotlagh**[3], **Varun Jampani**[4], **Guha Balakrishnan**[1]

[1]Rice University
[2]Independent Researcher
[3]The University of Queensland
[4]Arcade AI

## Abstract

Modern monocular 3D reconstruction methods and vision-language models (VLMs) demonstrate impressive results on standard benchmarks, yet recent works cast doubt on their true understanding of geometric properties. We introduce GIQ, a comprehensive benchmark specifically designed to evaluate the geometric reasoning capabilities of vision and vision-language foundation models. GIQ comprises synthetic and real-world images and corresponding 3D meshes of diverse polyhedra covering varying levels of complexity and symmetry, from Platonic, Archimedean, Johnson, and Catalan solids to stellations and compound shapes. Through systematic experiments involving monocular 3D reconstruction, 3D symmetry detection, mental rotation tests, and zero-shot shape classification tasks, we reveal significant shortcomings in current models. State-of-the-art reconstruction algorithms trained on extensive 3D datasets struggle to reconstruct even basic geometric Platonic solids accurately. Next, although foundation models may be shown via linear and nonlinear probing to capture specific 3D symmetry elements, they falter significantly in tasks requiring detailed geometric differentiation, such as mental rotation. Moreover, advanced vision-language assistants such as ChatGPT, Gemini and Claud exhibit remarkably low accuracy in interpreting basic shape properties such as face geometry, convexity, and compound structures of complex polyhedra. GIQ is publicly available at `toomanymatts.github.io/giq-benchmark/`, providing a structured platform to benchmark critical gaps in geometric intelligence and facilitate future progress in robust, geometry-aware representation learning.

## 1 Introduction

Computer vision models trained on massive image datasets now achieve state-of-the-art performance on a range of visual reasoning tasks. However, unlike humans who naturally reason about the world in terms of 3D structure, these models learn to exploit arbitrary (often 3D-unaware) patterns discovered from their training datasets. This property can, in turn, lead to poor generalization performance on out-of-distribution data. For example, vision-language models (VLMs) are known to struggle with questions related to depth ordering (Tong et al., 2024), and as shown in our experiments, monocular reconstruction algorithms struggle to reconstruct shapes outside of their training distributions. For this reason, there is growing interest in understanding the 3D "awareness" of popular foundation vision models, such as by "linearly probing" (Alain & Bengio, 2017) their intermediate features to solve basic 3D reasoning tasks such as depth estimation and scene registration (El Banani et al., 2024; Man et al., 2024).

These few existing 3D analysis studies rely on large datasets of synthetic or in-the-wild objects such as OmniObject3D (Wu et al., 2023), NAVI (Jampani et al., 2023), and Google Scanned Objects (GSO) (Downs et al., 2022). While these datasets enable large-scale analyses, they are not conducive to careful evaluation of visual reasoning with respect to core 3D object geometry properties such

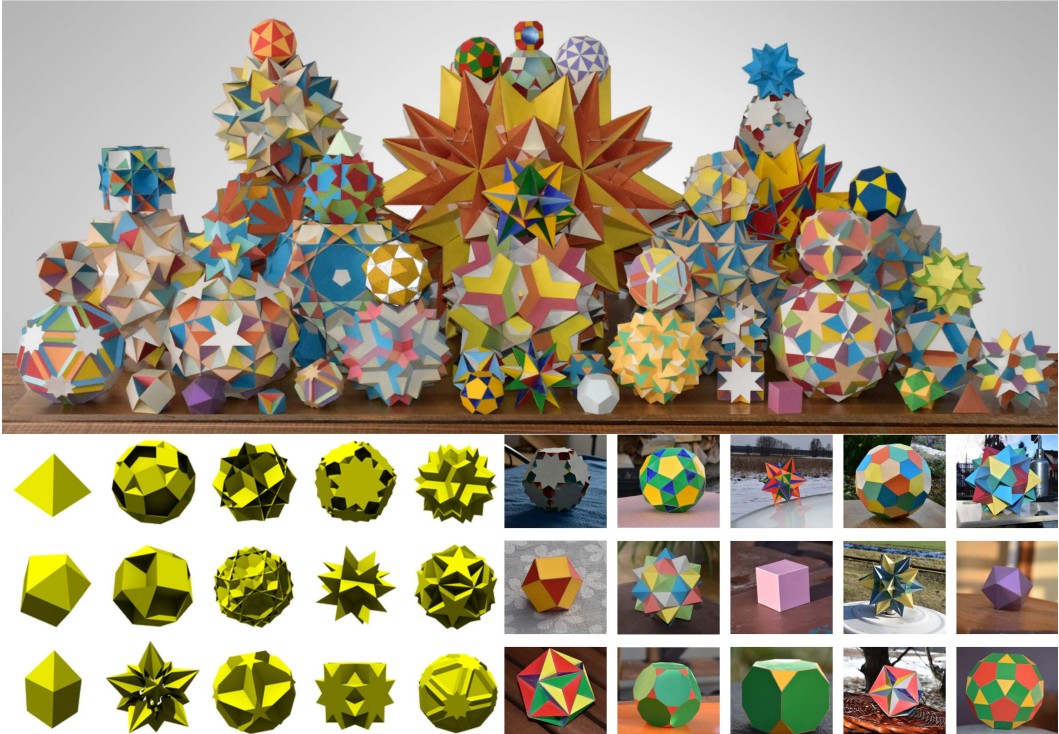

Figure 1: **Samples of synthetic and real 3D solids from our GIQ dataset.** A subset of the 224 real polyhedra included in our dataset, illustrating their variety in complexity, class, and colors. (bottom left) Simulated solids from Mitsuba Physically Based Renderer. (bottom right) Real polyhedra constructed from paper, placed in different realistic backgrounds.

as symmetry, convexity, and complexity. For example, is CLIP (Radford et al., 2021) better than DINO (Oquab et al., 2023) at identifying center-point symmetries, and if so, to what degree? And how do reconstructions of SF3D (Boss et al., 2024) and OpenLRM (Hong et al., 2023) compare as shape complexity increases? Quantitative answers to such questions can provide crucial information to benchmark and ultimately improve 3D geometric reasoning capabilities of these algorithms.

In this study, we introduce GIQ (for *Geometric IQ* Test), a first-of-its-kind dataset consisting of simulated and physical polyhedra to help answer such questions (see Fig. 1). Polyhedra have fascinated mathematicians for centuries due to their fundamental nature in geometry, inspiring classical works such as Plato's association of regular solids with classical elements, Kepler's exploration of polyhedral properties in his *Harmonices Mundi* (Kepler, 1619), and Euler's groundbreaking analysis of polyhedral topology (Euler, 1758). Polyhedra also appear in many practical scientific applications, from crystalline minerals (e.g., cubic iron pyrite) and molecular structures in nature (e.g., the truncated icosahedral "buckyball") to fundamental primitives in graphics algorithms.

Unlike arbitrary shapes, the well-defined categories of polyhedra (e.g., Platonic, Archimedean, Johnson solids) along with their precise symmetry groups (e.g., tetrahedral, octahedral, icosahedral) provide unambiguous ground truth for evaluation. Furthermore, polyhedra exhibit a rich hierarchy of geometric complexity – from simple Archimedean solids such as the tetrahedron with four faces to the highly complex great dirhombicosidodecahedron, nicknamed *Miller's Monster* (Verheyen, 1989), a nonconvex uniform polyhedron with 124 faces (40 triangles, 60 squares, and 24 pentagrams) – enabling systematic investigation of how shape complexity affects perception. The perceptual salience of these patterns makes polyhedra an excellent testbed for vision models.

To this end, we constructed GIQ with images of 224 unique polyhedra captured from multiple viewpoints. We constructed synthetic polyhedra using the Mitsuba Physically Based Renderer (Nimier-David et al., 2019) (see Fig. 1 bottom left), and constructed intricate physical polyhedra models from paper and placed them in various indoor and outdoor real-world environments (see

Fig. 1 bottom right). This is the first dataset of its kind with such a quantity and range of synthetic and real polyhedra.

We used GIQ to perform systematic evaluations on a range of state-of-the-art vision-language models and monocular 3D reconstruction methods to assess their capability to recognize symmetry, reconstruct complex geometries from a single view, and accurately reason about shape equivalence across diverse viewpoints and real-world conditions. We find that even for the simplest Platonic solids, single-image reconstruction is unreliable. Across state-of-the-art monocular 3D reconstruction methods trained on millions of diverse 3D assets, we observe frequent failures to recover even the properties of a cube: axis-aligned faces, right angles, uniform edge lengths, and underlying 3D symmetries (e.g., 4-fold rotational invariances), leading to asymmetric artifacts and mismatched global structure across views. Furthermore, as geometric complexity increases to nonconvex or compound forms, reconstruction quality degrades further, exhibiting fractured surfaces and topological inconsistencies. Interestingly, while we found that a lightweight non-linear probe on foundation image encoders (e.g., SigLip, DINOv2) can reliably classify symmetry cues, we also observed that these encoders and frontier vision–language assistants (ChatGPT, Gemini, Claude) struggle in classifying geometrically similar shapes. Finally, in zero-shot shape classification, vision-language assistants achieve below 20% accuracy on simple Catalan and Johnson solids and on non-convex shapes, frequently misidentifying face geometry, confusing convexity, or conflating compounds.

Results show that GIQ provides a targeted benchmark to diagnose fundamental geometric intelligence gaps in vision systems, laying the groundwork for future principled improvements in spatial perception and 3D-aware visual reasoning.

## 2 RELATED WORK

Foundation vision models, such as CLIP (Radford et al., 2021), DINO (Oquab et al., 2023), and Llama (Touvron et al., 2023), have achieved widespread popularity due to their remarkable performance across diverse vision tasks. Given their widespread adoption, recent studies increasingly investigate these models' robustness and generalization properties, including sensitivity to adversarial perturbations (Schlarmann & Hein, 2023), reliance on dataset biases (Nguyen et al., 2022), capacity for compositional reasoning (Doveh et al., 2023; Lewis et al., 2022), and resilience to visual anomalies (Zhu et al., 2024b). Among these efforts, there is a growing emphasis on evaluating how effectively these models encode and reason about 3D structures (El Banani et al., 2024; Man et al., 2024), spatial relations (Ramakrishnan et al., 2024; Tang et al., 2024; Yamada et al., 2023; Yang et al., 2023), and spatial cognition more broadly.

A particularly critical aspect of spatial cognition studied extensively in cognitive science is the ability to recognize objects under rotation. The Mental Rotation Test (MRT), first proposed by Shepard & Metzler (1971) and later standardized by Vandenberg & Kuse (1978), assesses spatial reasoning by requiring subjects to determine whether two rotated 3D objects are identical. Recent work by Ramakrishnan et al. (2024) examined mental rotation using synthetic stimuli consisting of Lego-like blocks. Building upon this, our study expands the MRT evaluation framework by incorporating polyhedral shapes of varying complexity and geometric properties, evaluated across both synthetic renderings and real-world images.

Another fundamental aspect of spatial intelligence extensively studied in cognitive science is symmetry perception. Humans exhibit a well-documented sensitivity and preference for symmetric patterns (Wagemans, 1997). In computer vision, symmetry detection has been explored both in two-dimensional contexts (Tsogkas & Kokkinos, 2012; Funk & Liu, 2017) and, more recently, extended into three-dimensional settings (Gao et al., 2020). Despite substantial advancements, existing large-scale 3D datasets like Objaverse (Deitke et al., 2023b) and its successor Objaverse XL (Deitke et al., 2023a) typically lack detailed annotations necessary to evaluate fine-grained geometric reasoning, such as recognition of specific symmetry groups or subtle shape complexities. Addressing this gap, we introduce a targeted polyhedron-based benchmark, providing clearly defined ground truths on symmetry, complexity, and geometric properties, thereby enabling precise measurement of geometric understanding in foundation vision models.

The choice of polyhedra as our evaluation focus draws on a rich heritage of mathematical and scientific exploration. Historically, scholars such as Da Vinci, Descartes, Euler, and Gauss extensively

| Group [#] | Examples | Group [#] | Examples |
|---|---|---|---|
| Platonic [5] |  | Archimedean [13] |  |
| Catalan [13] |  | Johnson [92] |  |
| Stellations [48] |  | Kepler-Poinsot [4] |  |
| Compounds [10] |  | Uniform non-convex [53] |  |

Figure 2: **Summary of polyhedral groups in GIQ, highlighting group names, counts of distinct 3D shapes (in parentheses), and representative examples.** Platonic, Archimedean, and Catalan solids are convex, while Kepler-Poinsot polyhedra and compounds represent special cases of stellations; consequently, the sum of group counts (238) exceeds the 224 unique shapes in the dataset. The categorization presented here is arbitrary: polyhedra possess numerous properties allowing various groupings; we selected this set as a representative example.

explored these geometric forms, recognizing their structural elegance and conceptual clarity. In modern mathematics, polyhedra continue to be studied rigorously by scholars (Coxeter et al., 2012; 1954; Klein, 2003; Stewart, 1980). Building on this historical foundation, our dataset systematically leverages polyhedral geometry to rigorously probe and enhance the geometric intelligence of modern vision systems.

## 3 DATASET

We describe the composition of GIQ in this section. We first present geometrical concept definitions in Sec. 3.1, followed by a description of all polyhedral shapes we considered in Sec. 3.2. We then describe the synthetic and real instantiations of these shapes in GIQ in Sec. 3.3 and Sec. 3.4.

### 3.1 GEOMETRY PRELIMINARIES

We first briefly define key geometric concepts related to polyhedra which will underpin the detailed discussion of polyhedral classes presented in the following subsections. Following Coxeter's definition, a polyhedron is a finite set of polygons arranged so that every side of each polygon belongs to exactly one other polygon, with no subset of polygons having this property (i.e. the entire collection is connected) (Coxeter et al., 1954). A *regular polygon* is equilateral (all sides equal) and equiangular (all internal angles equal). A polyhedron with faces that are all congruent regular polygons (i.e., identical in size and shape) is termed a **regular polyhedron**. A polyhedron is **convex** if any line segment joining two points inside or on its surface remains completely within or on its surface. Otherwise, it is considered **non-convex** (or concave). A polyhedron is **vertex-transitive** if any vertex can be mapped onto any other vertex by a symmetry operation (rotation, reflection, or translation). Similarly, a polyhedron is **edge-transitive/face-transitive** if all edges/faces can be mapped onto each other by symmetry operations. A **uniform polyhedron** has regular polygonal faces and is vertex-transitive, meaning that the arrangement of its faces around each vertex is identical. Lastly, **dual polyhedra** are pairs of polyhedra where vertices and faces are interchanged: vertices of one polyhedron correspond exactly to faces of the other, and vice versa.

### 3.2 3D SHAPES

We considered a comprehensive collection of polyhedral classes systematically varying in geometric complexity, symmetry properties, and topological regularity. We began with the *Platonic solids*, comprising exactly five regular convex polyhedra: tetrahedron, cube, octahedron, dodecahedron, and icosahedron. Relaxing the condition that all faces must be identical but retaining vertex-transitivity

and edge-transitivity yields the broader set of *Archimedean solids*, consisting of 13 convex polyhedra whose faces are regular polygons of more than one type, arranged symmetrically around each vertex. Taking the dual polyhedra of the Archimedean solids leads to the 13 *Catalan solids* (Catalan, 1865). Each Catalan solid has congruent faces but allows variation in vertex configurations. Removing the requirement for vertex and face transitivity entirely produces the family of *Johnson solids*, comprising 92 strictly convex polyhedra with regular polygonal faces but lacking vertex uniformity (Johnson, 1966; Zalgaller, 1969). To further enrich the diversity of shapes in our dataset, we incorporated *stellations*, geometric constructions formed by extending the faces or edges of polyhedra until they intersect again, creating more complex, nonconvex forms. We included selected stellations of the octahedron, dodecahedron, icosahedron, cuboctahedron, and icosidodecahedron (Coxeter et al., 2012). Additionally, we included various *compound polyhedra*: structures formed by the symmetric combination of multiple polyhedra (e.g., compound of cube and octahedron, and compound of dodecahedron and icosahedron). Both stellations and compound polyhedra significantly enhance structural complexity, introducing intricate symmetry properties and challenging visual configurations. Finally, we considered the complete set of polyhedra described by Wenninger (1971), encompassing 119 shapes in total. Beyond Platonic, Archimedean, and stellations, this set includes the *Kepler-Poinsot solids*, and a collection of nonconvex polyhedra exhibiting a wide array of geometric and topological complexities.

In total, the GIQ dataset consists of 224 carefully curated 3D shapes, from symmetric and simple regular forms to intricate, nonconvex, and irregular structures, providing a robust basis for exploring geometric perception and spatial reasoning in neural models. We summarize counts of specific shapes in the dataset in Figure 2 with additional representative samples and listed key geometric features for each group provided in the appendix.

## 3.3 SYNTHETIC RENDERINGS

To systematically evaluate spatial reasoning under controlled conditions, we rendered each of the 3D shapes described above using the Mitsuba physically-based renderer Nimier-David et al. (2019) from 20 randomly sampled viewpoints. We distributed these viewpoints uniformly over the viewing hemisphere, ensuring diverse perspective coverage. We used a perspective camera with a resolution of $256 \times 256$ pixels, a near clipping plane of $10^{-3}$, and a far clipping plane of $10^8$. For each view, the object is rendered using diffuse shading. To simulate realistic lighting while preserving shape detail, we used a two-sided diffuse BRDF with a high-reflectance yellowish surface. Each rendering uses 1024 low-discrepancy samples per pixel to ensure smooth convergence. Instead of global illumination, which tended to reduce contrast between adjacent faces and obscure geometric boundaries, we adopted a direct integrator with four emitter samples and no BSDF sampling. This setup emphasizes sharp direct shading cues and produces face-to-face contrast more consistent with wild images, where distinguishing neighboring facets is typically easier.

## 3.4 WILD IMAGES

To complement our synthetic renderings with real-world variabilities, we placed physical paper models of the polyhedra—constructed by Piotr Pawlikowski—in natural settings. Our wild image dataset includes all Platonic, Archimedean, Catalan, and other non-prismatic nonconvex uniform polyhedra, as well as the Kepler–Poinsot solids and a representative subset of 48 Johnson solids. These paper models were manually assembled following geometric blueprints.

We photographed each shape using a Nikon D3500 DSLR camera at a native resolution of $6000 \times 4000$ pixels under two broad conditions. First, we captured approximately 20 indoor images per shape under controlled lighting. Second, we collected 20 outdoor images per shape under varying environmental conditions, including sunny weather, overcast skies, and snowy winter backgrounds. These outdoor images introduce rich natural variability in illumination, background texture, and color. We randomly selected viewpoints to ensure full 360-degree coverage, while maintaining sufficient visibility of shape geometry. We provide a visual overview of our wild image dataset in Figure 1; additional examples are provided in the appendix.

## 4 EXPERIMENTS

We used the GIQ dataset to conducted a series of comprehensive experiments to evaluate the geometric reasoning capabilities of contemporary foundation vision models. We focused on four types of experiments: *Monocular 3D Reconstruction*, *3D Symmetry Detection*, *Mental Rotation Tests*, and *Zero-Shot Polyhedron Classification*. These experiments are designed to probe different dimensions of geometric intelligence: explicit 3D reconstruction of shapes from single images, implicit embedding-based detection of 3D symmetries and subtle geometric distinctions, and high-level semantic classification of frontier vision-language models. We note that while "wild" images retained their complex backgrounds and varying lighting conditions for Symmetry, Mental Rotation, and Classification tasks to test environmental robustness, they were preprocessed (background removal and centering) exclusively for the Monocular 3D Reconstruction task to align with the input assumptions of baseline methods. Detailed quantitative results, separated by synthetic and wild domains, are provided in the appendix. Collectively, these evaluations highlight both the current strengths and shortcomings of state-of-the-art models to represent, recognize, and reason about complex geometric structures.

### 4.1 MONOCULAR 3D RECONSTRUCTION

We first evaluated three recent state-of-the-art monocular 3D reconstruction methods: **Shap-E** (Jun & Nichol, 2023), **Stable Fast 3D** (Boss et al., 2025), and **OpenLRM** (He & Wang, 2023). Shap-E is a diffusion-based model trained on millions of 3D assets. It encodes shapes with a Transformer and generates textured meshes via implicit neural representations. Stable Fast 3D is built on TripoSR (Tochilkin et al., 2024), encodes images with DINOv2 (Oquab et al., 2023) and outputs triplane-based representations. It is trained on a curated subset of Objaverse (Deitke et al., 2023b). OpenLRM, based on LRM (Hong et al., 2024), predicts neural radiance fields from single images using a transformer, and is trained end-to-end on Objaverse and MVImgNet (Yu et al., 2023). We evaluated these pre-trained models zero-shot, using GIQ as an out-of-distribution test set.

The experimental task involves reconstructing the complete 3D geometry of polyhedral shapes from a single image. We selected representative examples across three categories: cube (Platonic solid), great dodecahedron (Kepler-Poinsot solid), and small cubicuboctahedron (uniform nonconvex solid). To provide a fair comparison, we preprocessed wild images via center cropping and background removal. Table 1 presents results of this evaluation. Despite extensive training on millions of diverse shapes or inclusion of similar solids in the training dataset, all three models exhibited significant reconstruction failures for most tested cases, particularly on wild images and shapes beyond the simplest forms. Shap-E successfully reconstructed the synthetic cube but failed dramatically on the wild cube and more complex shapes. Stable Fast 3D accurately captured front-facing geometries for several shapes but consistently failed to coherently reconstruct side and rear views. Similarly, OpenLRM produced plausible reconstructions for simple synthetic shapes but exhibited considerable inaccuracies across viewpoints and struggled substantially with complex and wild imagery. Additional qualitative and quantitative reconstruction results are provided in the appendix.

### 4.2 3D SYMMETRY DETECTION

We evaluated the capability of various image encoders to detect specific 3D symmetry elements from single-image inputs. The task involved predicting the presence of three distinct symmetry elements in objects depicted by images: central point reflection (invariance under inversion through a central point), and $n$-fold rotational symmetries, defined as invariance under rotation by $360°/n$. Specifically, we target 4-fold ($90°$) and 5-fold ($72°$) symmetries. . Unlike common evaluations focused on 2D planar symmetries or image rotations, our evaluation explicitly targets recognition of inherent symmetry elements defined with respect to the object's three-dimensional structure.

We examined a diverse set of encoders with variations in modalities, architectures, and supervision strategies. Specifically, we selected six supervised methods, three image-and-text transformer-based models (CLIP (Radford et al., 2021), DreamSim (Fu et al., 2023), SigLip (Zhai et al., 2023)), and three image CNN-based ones (DeiT III (Touvron et al., 2022), SAM (Foret et al., 2020; Chen et al., 2021), and ConvNext (Liu et al., 2022)). Additionally, we considered three self-supervised transformer-based methods trained solely on images: DINO (Caron et al., 2021), DINOv2 (Oquab et al., 2023), and

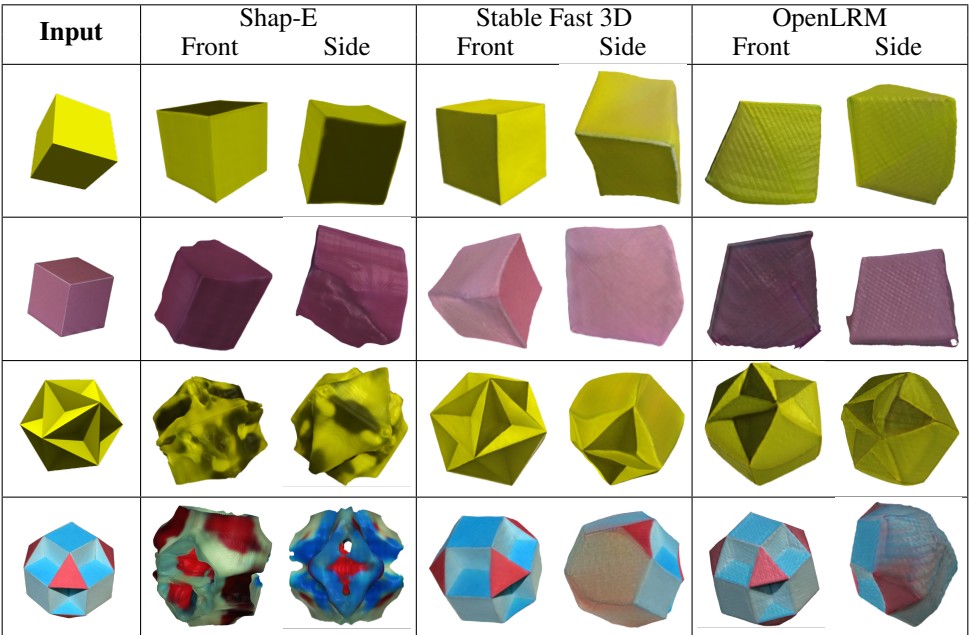

| **Input** | Shap-E | | Stable Fast 3D | | OpenLRM | |
| --- | --- | --- | --- | --- | --- | --- |
| | Front | Side | Front | Side | Front | Side |

Table 1: Monocular 3D reconstruction results. Each method reconstructs a 3D shape from the input image, which we visualize by rendering the output from selected viewpoints. Rows depict pairs of synthetic and wild images of a cube (platonic solid), great dodecahedron (Kepler-Poinsot solid), and small cubicuboctahedron (uniform nonconvex solid).

Masked AutoEncoder (MAE) (He et al., 2022). Finally, to assess geometry-native representations, we included three multi-view pretrained networks: VGGT (Wang et al., 2025), DUSt3R (Wang et al., 2024), and MASt3R (Leroy et al., 2024).

We applied a linear probe (Alain & Bengio, 2017), feeding each featurizer's embeddings into a single linear layer to classify symmetry elements. To assess if geometric information was encoded in non-linearly separable manifolds, we also evaluated non-linear probes; however, these yielded comparable average performance to linear probes (see appendix). To address class imbalance, we employed a weighted binary cross-entropy loss. For each symmetry class $c$, we compute the positive-class weight as $w_c = \frac{N - n_c}{n_c}$, where $n_c$ is the number of positive samples for class $c$ and $N$ is the total number of samples. We then define the per-example loss for logits $z_{i,c}$ and targets $y_{i,c} \in \{0, 1\}$ as:

$$\mathcal{L} = -\frac{1}{N} \sum_{i=1}^{N} \sum_{c=1}^{C} \left[ w_c \, y_{i,c} \log \sigma(z_{i,c}) + (1 - y_{i,c}) \log(1 - \sigma(z_{i,c})) \right],$$

where $\sigma(z_{i,c})$ denotes the predicted probability for class $c$ on example $i$. We used balanced accuracy, computed as $0.5 \cdot \frac{\text{TP}}{P} + 0.5 \cdot \frac{\text{TN}}{N}$, where TP and TN represent true positives and true negatives respectively (with $P$ and $N$ being the number of positive and negative samples), as our primary evaluation metric.

We trained models on embeddings from synthetic images and subsequently evaluated them on embeddings extracted from both synthetic and real-world (wild) images. We excluded Johnson solids due to ambiguities arising from certain viewing angles. To ensure robust performance estimates and mitigate potential biases from specific dataset splits, we employed a 5-fold cross-validation strategy. We partitioned the unique polyhedral shapes into five disjoint folds, ensuring that in every iteration, the test set contained only shapes unseen during training. Detailed information about dataset composition is provided in the appendix.

We present the averaged balanced accuracy across all five folds in Figure 3 (left). Notably, DINOv2 consistently delivered strong performance across symmetry categories and particularly excelled in the recognition of 4-fold rotational symmetry, achieving up to 93% accuracy on wild images despite being trained only on synthetic data.

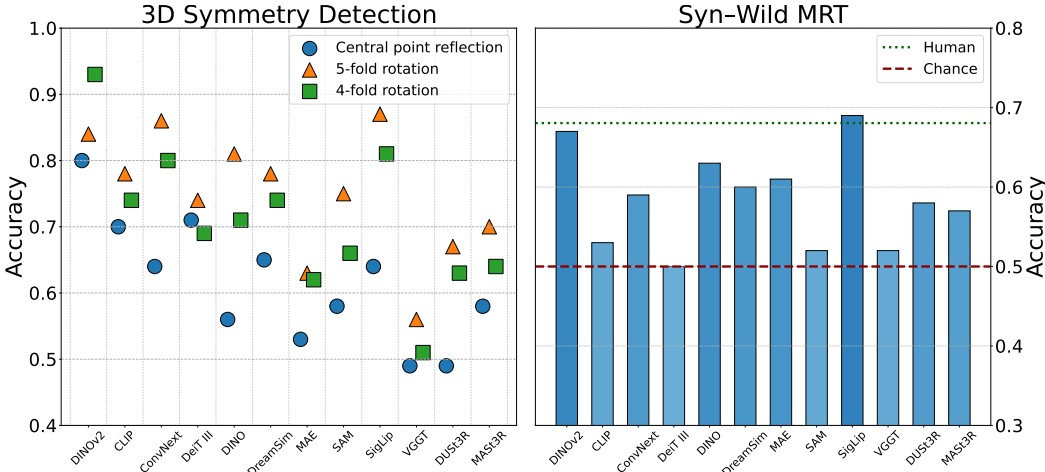

Figure 3: Left: Balanced accuracy ($0.5 \cdot \frac{TP}{P} + 0.5 \cdot \frac{TN}{N}$) for linear probing of 3D symmetry detection using embeddings from different featurizers. The linear classifier is trained only on synthetic images (Syn), and evaluated on real-world (Wild) images for detecting three symmetry types: central point reflection, 5-fold rotation, and 4-fold rotation. Right: Mental Rotation Test accuracy using non-linear probes. Top models (e.g., SigLIP) match the human average (~69%, green dotted line), though 68% of human participants still outperformed the best model.

### 4.3 MENTAL ROTATION TEST

Next, we evaluated foundation image encoders' spatial reasoning capabilities through a Mental Rotation Test (MRT) (Shepard & Metzler, 1971). This task involves determining whether two images-one synthetic rendering and one real-world (wild) photograph-depict the same polyhedral object, differing only by rotation. Such a scenario is relevant to real-world applications, for instance, a robot trained on CAD models needing to locate a physical object with matching geometric properties in an unstructured environment.

We first used an 80%-20% train-test split with synthetic image pairs alone to establish baseline performance. Under these simplified conditions, model accuracies ranged between 93%-98% (complete results available in appendix). Through this initial experiment, we identified that combining image embeddings using absolute difference ($|e_1 - e_2|$) followed by a non-linear probe yielded the highest accuracy. We further explored alternative embedding combination methods (concatenation and subtraction) and linear probing baselines with detailed results per featurizer reported in the appendix However, since the trivial setting posed insufficient challenges for discerning nuanced differences, we subsequently introduced a more demanding *hard* split. This split is specifically designed with visually and geometrically similar polyhedra pairs, rigorously testing the models' fine-grained shape differentiation abilities. Examples of these challenging shape pairs are shown in the appendix.

Results for the *hard* split are summarized in Figure 3 (right). We trained on synthetic images and evaluated on both synthetic and synthetic-wild pairs, with syn-syn results provided in the appendix. All models exhibited a significant performance drop compared to the trivial split, particularly in the synthetic-wild test scenario, where average performance approached chance level. While non-linear probes enabled SigLIP and DINOv2 to achieve 69% and 67% accuracy respectively, the majority of models exhibited limited spatial reasoning capabilities, often struggling to surpass 60%, highlighting the considerable difficulty models face in reliably distinguishing subtle geometric differences.

To determine the extent to which these geometric cues are perceptible to human observers versus current vision models, we established a human baseline. We conducted a user study with 42 participants using a web interface to evaluate the exact image pairs from our test set. Participants answered 25 questions (5 from the "easy" split and 20 from the "hard" split) with a strict 30-second time limit per question. On the easy set, participants achieved a mean accuracy of 97.56%, confirming task comprehension. On the hard set, human accuracy averaged 68.05% (std dev: 0.11), with top performers scoring 90% (18/20). Notably, while our best-performing model configuration

(SigLIP with a non-linear probe) achieved parity with the average human (∼69%), 68% of individual participants outperformed the best model.

### 4.4 Zero-Shot Polyhedron Classification by Frontier Models

Finally, we conducted a zero-shot polyhedron classification task to evaluate the geometric reasoning capabilities of frontier vision-language models. Specifically, we assessed Claude 3.7 Sonnet (Anthropic, 2025), Gemini 2.5 Pro (DeepMind, 2025), ChatGPT o3, and ChatGPT o4-mini-high (OpenAI, 2025) by querying each model with synthetic and real-world images from our dataset with the prompt: *"What is the name of this polyhedron?"*. To ensure robust evaluation, we accounted for synonyms (e.g., Cube/Hexahedron, Stella Octangula/Stellated Octahedron) and manually verified that no models produced correct answers using synonyms outside our dictionary. Classification accuracy across polyhedron categories is reported in Figure 4 (a).

Model performance varied significantly across polyhedron categories. Gemini 2.5 Pro achieved perfect accuracy for Kepler-Poinsot solids, while ChatGPT o3 achieved a perfect score on Platonic solids. Conversely, all models struggled with Johnson solids, Catalan solids, uniform non-convex solids, and compound structures, indicating that even shapes with simple repetitive polygonal faces (e.g., Johnson or Catalan) remain difficult to classify, alongside the more complex non-convex cases. To investigate potential bottlenecks in prompting or input ambiguity, we extended our evaluation to include Chain-of-Thought (CoT) prompting and Multi-View (MV) inputs (providing three canonical views). CoT strategies yielded minimal gains, with models often hallucinating intermediate features. Similarly, MV inputs provided only marginal improvements, primarily for low-symmetry Johnson solids where single views can be ambiguous. For high-symmetry categories (Platonic, Archimedean, Compounds, Stellations), a single view is informationally complete; thus, the persistent failure indicates a lack of geometric reasoning rather than visual occlusion. Detailed results for these ablations are provided in the appendix.

We qualitatively analyzed classification errors, revealing systematic patterns of geometric misinterpretation, and present input images with frontier-model reasoning in Figure 4 (b); additional examples appear in the appendix. Models frequently (i) confused convex and concave structures—for example, **Claude 3.7 Sonnet** correctly described face types and coloring but still misclassified a concave shape as a cuboctahedron, an Archimedean solid that must be convex; (ii) conflated compounds with stellations or other multi-component assemblies—for instance, **ChatGPT o3** recognized a star-shaped compound but misidentified both the protrusions (triangular pyramids vs. octahedra) and the constituents (dodecahedron–icosahedron vs. a compound of octahedra), while **Gemini 2.5 Pro** similarly misclassified a color-coded cube–octahedron compound as a "stellated octahedron"; and (iii) hallucinated additional face types, as seen when **ChatGPT o4-mini-high** noted pentagonal faces but incorrectly added hexagons, ultimately blaming viewpoint ambiguity despite the full pattern being visible.

To determine if these limitations were specific to 2D-pretrained architectures, we extended our evaluation to geometry-native models. We tested LLaVA-3D Zhu et al. (2024a) (using image inputs), as well as ShapeLLM Qi et al. (2024) and PointBind & PointLLM Guo et al. (2023) (using ground truth point clouds as input). As shown in Figure 4 (a), despite the availability of explicit 3D geometry, these models did not outperform the generalist frontier VLMs. Collectively, these limitations underscore a fundamental gap in current architectures, highlighting the necessity of enhanced mechanisms for explicit geometric representation to reliably differentiate complex structures.

## 5 Conclusion

We introduced GIQ, a novel benchmark dataset designed to systematically assess the geometric reasoning capabilities of contemporary foundation vision and vision-language models using both synthetic and real-world polyhedral structures. Our extensive evaluations reveal notable discrepancies in current state-of-the-art models, highlighting a substantial disconnect between their impressive capabilities on standard benchmarks and their performance in tasks requiring explicit geometric reasoning.

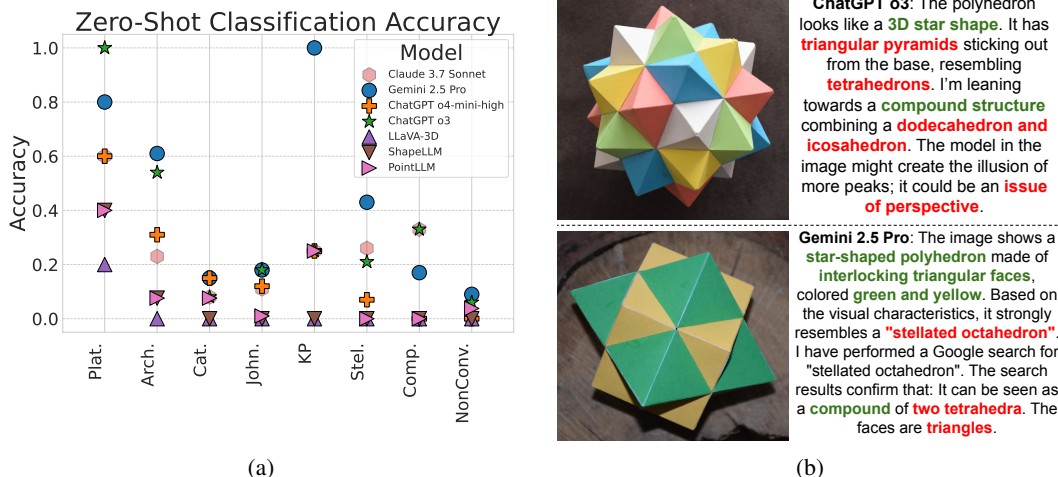

Figure 4: (a) Zero-shot classification accuracy of various frontier models across polyhedron categories using wild images. Results on synthetic images showed only marginal differences and are provided in the appendix. (b) Qualitative reasoning failures of frontier vision-language models. Correct text in **green**, incorrect in **red**.

The results present a nuanced picture. On one hand, our 3D symmetry detection experiments show that pretrained vision encoders like DINOv2 can be surprisingly effective, as their image embeddings inherently capture fundamental 3D structural properties without explicit 3D supervision. This demonstrates the potential of foundation models to implicitly encode 3D structure and opens the door to utilizing them as lightweight, plug-and-play modules in downstream tasks. On the other hand, this implicit understanding does not translate to robust, explicit reasoning in other domains. Our experiments underscore critical limitations in current models. Monocular 3D reconstruction approaches consistently failed to capture fundamental geometric properties from single images, even when trained on extensive, diverse datasets containing similar geometric primitives. Likely, these models learn priors for noisy, imperfect surfaces from their training data rather than mathematical exactness, suggesting that scale alone is insufficient without mathematically generated data to enforce geometric constraints. Similarly, mental rotation tests revealed an inability to capture subtle geometric distinctions. While the strongest model achieved parity with the average human baseline using non-linear probe, the vast majority of architectures faltered significantly. Crucially, our user study exposes a persistent gap in robust reasoning: a substantial majority of human participants outperformed even the best model, with top subjects demonstrating a decisive advantage on complex geometric instances. This was further confirmed by our zero-shot classification experiments, which revealed systematic errors in frontier models; they frequently misidentified fundamental properties such as face geometry, convexity, and compound structures, particularly within complex non-convex polyhedral classes.

We view GIQ as a "geometric litmus test" for spatial intelligence. Because geometric primitives are the building blocks of complex reasoning, failure here implies a fragile, texture-based approximation rather than true understanding. Real-world tasks rely on this same "geometric arithmetic"; thus, models failing GIQ are unprepared for the "geometric calculus" of complex scenes. While establishing a precise correlation with arbitrary organic shapes requires further empirical study, GIQ isolates necessary foundational skills, paving the way for future rigorous evaluations of both rigid and fluid dynamics.

Looking forward, GIQ offers a promising pathway for improving geometric intelligence. Its structured ontology enables the automatic generation of precise Chain-of-Thought (CoT) training data for step-by-step reasoning, though future training-focused extensions must scale to broader wild distributions to ensure robust generalization. Ultimately, GIQ serves as a critical diagnostic tool, providing the framework needed to drive the development of robust, geometry-aware foundation models capable of human-level spatial reasoning.

**Ethics statement:** Authors used LLMs to help improve grammar and wording during the preparation of this manuscript.

**Acknowledgement.** Supported by the Intelligence Advanced Research Projects Activity (IARPA) via Department of Interior/ Interior Business Center (DOI/IBC) contract number 140D0423C0076. The U.S. Government is authorized to reproduce and distribute reprints for Governmental purposes notwithstanding any copyright annotation thereon. Disclaimer: The views and conclusions contained herein are those of the authors and should not be interpreted as necessarily representing the official policies or endorsements, either expressed or implied, of IARPA, DOI/IBC, or the U.S. Government.

We thank the students from Adam Mickiewicz High School in Kluczbork, Poland, for their valuable contribution in constructing the Johnson solids used in this work.

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
