# GIQ: Benchmarking 3D Geometric Reasoning of Vision Foundation Models with Simulated and Real Polyhedra
## Supplementary Material

This supplementary material provides additional details and extended experimental analyses to complement and further support the results presented in the main paper. Specifically, it includes:

- Extended summary of polyhedral groups: their key geometric features, and additional representative samples.

- Extended qualitative and quantitative evaluations of monocular 3D reconstruction methods across various polyhedral categories.

- Detailed statistics and descriptions of dataset splits for the 3D symmetry detection task, along with synthetic-to-wild generalization gaps across featurizers and symmetry types.

- Additional analyses and results from the Mental Rotation Test (MRT), covering both trivial and challenging setups, along with example test pairs from the hard split.

- Expanded zero-shot polyhedron classification results across synthetic and real-world images, with additional qualitative examples highlighting common reasoning failures.

- Extended probing analysis comparing linear versus non-linear (MLP) heads for MRT and Symmetry Detection, including results for three additional multi-view pretrained models (VGGT, DUSt3R, MASt3R).

- Ablation studies on zero-shot classification, evaluating 3D-native VLMs (LLaVA-3D, ShapeLLM, PointLLM) and the impact of Multi-View inputs and Chain-of-Thought prompting strategies.

- Additional qualitative samples from the "Wild" dataset partition, illustrating the diversity of indoor and outdoor environmental conditions.

## 1 Expanded Characterization of the Polyhedral Domain

To provide a more comprehensive understanding of the geometric domain covered by our GIQ dataset, we present in Table 5 an extended summary of the polyhedral groups used in our study. This table details families of polyhedra, from the well-known Platonic and Archimedean solids to more complex groups like the Johnson solids and non-convex stellations. For each group, we outline the defining geometric properties, specify the number of solids within that category, and provide additional representative visual examples. Beyond this taxonomy, polyhedra exhibit many further attributes—e.g., chirality, face-type distributions, stellation depth, or Rupert property—that can be leveraged to design additional benchmarks, enabling targeted evaluation of geometric reasoning across tasks such as duality reasoning, convexity discrimination, component decomposition, or face-type counting.

## 2 Extended Monocular 3D Reconstruction Analyses

This section provides additional quantitative and qualitative evaluations of state-of-the-art monocular 3D reconstruction methods, specifically Shap-E (Jun & Nichol, 2023), Stable Fast 3D (Boss et al., 2024), and OpenLRM (He & Wang, 2023), across diverse polyhedral categories. We qualitatively

evaluate reconstructions using standard geometric similarity metrics: F-score (Tatarchenko et al., 2019), Chamfer Distance (Borgefors, 1986), and Hausdorff Distance (Aspert et al., 2002), which jointly capture point-level accuracy and surface coverage. Results showed in Table 9 consistently reveals low F-scores (below 0.6) across all tested methods and categories, underscoring the limitations of current approaches in capturing the complex geometric intricacies inherent in polyhedral structures. Additionally, qualitative assessments for selected shapes from Archimedean, compound, and stellation groups (Table 8) highlight clear deficiencies in preserving critical geometric details, symmetry properties, and overall structural coherence, indicating significant scope for methodological advancements.

## 3 DATASET SPLITS AND COMPOSITION FOR 3D SYMMETRY DETECTION

We provide a detailed description of the dataset structure and splits used for the 3D symmetry detection experiments. To ensure robust evaluation, we employed a 5-fold cross-validation strategy. In each fold, the test split exclusively contains views from 26 unique polyhedral shapes not present in the training set, enabling rigorous evaluation of model generalization to unseen geometries. Table 6 summarizes the distribution of positive and negative examples and their ratios for central point reflection, 5-fold rotation, and 4-fold rotation symmetries for a representative fold (Fold 1).

Unlike the main paper, which reports only Wild performance, Table 7 reports balanced accuracies for linear probes trained on synthetic images and evaluated on both synthetic and Wild inputs for central point reflection, 5-fold rotation, and 4-fold rotation. As expected given the training domain, accuracies are higher on synthetic inputs, while the synthetic-to-Wild generalization gap depends on both featurizer and symmetry: for example, CLIP shows a minimal gap on 5-fold ($0.80 \rightarrow 0.78$), DINO suffers large drops—especially on 4-fold ($0.87 \rightarrow 0.61$) and also on 5-fold ($0.88 \rightarrow 0.71$) while DINOv2 generalizes strongly on 4-fold ($0.96 \rightarrow 0.93$) and ties for best Wild 5-fold ($0.85$).

## 4 EMBEDDING STRATEGIES AND ADDITIONAL MENTAL ROTATION RESULTS

We provide additional analyses for the Mental Rotation Test (MRT), reporting results for both simplified (trivial) and challenging (hard) experimental setups, with representative examples of the hard split shown in Table 12. Under the simplified scenario (Table 10), synthetic image pairs with an 80%-20% train-test split are used. Most models achieve high accuracy (93%–98%) when employing the absolute difference embedding method. In contrast, concatenation and raw subtraction embedding methods, which include randomized embedding ordering during training, yield near-random performance (50% accuracy). For the challenging scenario (Table 11), additional analyses using raw subtraction and concatenation embeddings further confirm their consistently inferior performance compared to the absolute difference embeddings.

## 5 ZERO-SHOT POLYHEDRON CLASSIFICATION: SYNTHETIC VS. REAL-WORLD

Finally, we provide expanded analyses for the zero-shot polyhedron classification experiments presented in the main paper. Table 13 compares classification performance of four leading vision-language models—ChatGPT o3, ChatGPT o4-mini-high, Gemini 2.5 Pro, and Claude 3.7 Sonnet—using both synthetic and real-world ("wild") images. Results indicate only marginal performance differences between synthetic and real-world inputs, confirming consistent capabilities across these domains. However, polyhedral categories such as Catalan solids, Johnson solids, compound structures, stellations and uniform non-convex polyhedra remain particularly challenging, underscoring persistent limitations in geometric reasoning within current frontier vision-language models.

To further illustrate these challenges, Figure 1 presents additional qualitative examples of the models' reasoning processes. These cases reveal a recurring failure mode: models correctly detect local cues (e.g., pentagonal faces and color/pattern) but miscompose them into the wrong global structure—hallucinating absent elements (hexagons), overlooking nonconvexity (e.g., labeling a noncon-

vex solid as Archimedean, though all Archimedean solids are convex), misreporting face types, and at times rationalizing the error by blaming viewpoint.

## 6 EXTENDED PROBING ANALYSIS: LINEAR VS. NON-LINEAR

In this section, we provide the complete results comparing linear versus non-linear (MLP) probing performance for both the Mental Rotation Test (MRT) and 3D Symmetry Detection tasks, as summarized in Table 1. Additionally, we include results for the newly added multi-view pretrained models: VGGT Wang et al. (2025), DUSt3R Wang et al. (2024), and MASt3R Leroy et al. (2024). Regarding the probing architectures, the linear probe consists of a single affine layer mapping the input embedding dimension $D$ to the target class dimension $C$. The non-linear probe is implemented as a Multi-Layer Perceptron (MLP) with one hidden layer. Specifically, it maps the input $D$ to a hidden dimension $H = \min(1024, D)$, applies a ReLU activation function, and projects the result to the output dimension $C$. To ensure adequate convergence for the increased parameter count, non-linear probes were trained for twice the number of epochs compared to the linear baselines. Our results indicate that while non-linear probes yield significant gains for the Mental Rotation Test (enabling SigLIP to reach 69% accuracy), they offer negligible improvements for Symmetry Detection, suggesting that symmetry cues are largely linearly separable. Furthermore, despite their native 3D training, multi-view models do not consistently outperform strong 2D baselines (e.g., DINOv2) on these discriminative geometric tasks.

## 7 ABLATION STUDIES: MULTI-VIEW, COT, AND 3D-NATIVE MODELS

Here we provide detailed quantitative results for our additional ablation studies, including the evaluation of 3D-native VLMs (LLaVA-3D Zhu et al. (2024), ShapeLLM Qi et al. (2024), PointLLM Guo et al. (2023)) and the comparison of Single-View (SV) vs. Multi-View (MV) inputs with Chain-of-Thought (CoT) prompting.

For the prompting experiments, we defined the specific queries as follows:

Baseline Prompt: "*What is the name of this polyhedron?*"

Chain-of-Thought (CoT) Prompt: "*Let's identify this polyhedron by thinking step-by-step: Convexity: First, analyze its overall shape. Is this a convex polyhedron or is it non-convex? Symmetry and Rotation: Second, describe its symmetries. What rotational symmetries does it have? Does it have planes of reflectional symmetry? Faces and Vertices: Third, describe its components. What is the shape of each face, and how many faces meet at each vertex? Conclusion: Based on this analysis of its convexity, symmetry, and components, what is the precise name of this polyhedron?*"

As detailed in Table 2, 3D-native models did not exhibit a significant advantage over 2D-pretrained VLMs. For instance, even with access to ground truth point clouds, PointLLM achieved only 40% accuracy on Platonic solids and failed to classify most complex categories (e.g., 0% on Stellations and Compounds). Furthermore, Table 3 indicates that Chain-of-Thought (CoT) prompting yielded negligible gains and often degraded performance due to hallucinations. Multi-view inputs provided a minor boost for low-symmetry shapes (e.g., Johnson solids), resolving projection ambiguities, but failed to improve recognition for high-symmetry classes where the failure stems from reasoning rather than viewpoint selection.

## 8 ADDITIONAL REAL-WORLD QUALITATIVE SAMPLES

To further illustrate the diversity and complexity of the real-world data distribution in GIQ, we provide expanded qualitative samples in Table 4. These images highlight the significant domain shift introduced by the "Wild" split, encompassing diverse indoor environments with artificial lighting (Rows 1–4) as well as outdoor settings featuring natural illumination, shadows, and varied backgrounds (Rows 5–9).

| Featurizer | Syn-Wild MRT | 4-fold rot. | 5-fold rot. | CPF |
|---|---|---|---|---|
| Linear probe | | | | |
| DINOv2 | 0.43 | **0.93** | 0.84 | **0.80** |
| SigLIP | 0.54 | 0.81 | **0.87** | 0.64 |
| ConvNeXt | **0.60** | 0.80 | 0.86 | 0.64 |
| CLIP | 0.39 | 0.74 | 0.78 | 0.70 |
| DreamSim | 0.49 | 0.74 | 0.78 | 0.65 |
| DeiT III | 0.41 | 0.69 | 0.74 | 0.71 |
| DINO | 0.54 | 0.71 | 0.81 | 0.56 |
| SAM | 0.42 | 0.66 | 0.75 | 0.58 |
| MASt3R | 0.48 | 0.64 | 0.70 | 0.58 |
| DUSt3R | 0.53 | 0.63 | 0.67 | 0.49 |
| MAE | 0.42 | 0.62 | 0.63 | 0.53 |
| VGGT | 0.47 | 0.51 | 0.56 | 0.49 |
| Mean (linear) | 0.478 | 0.69 | 0.75 | 0.62 |
| Nonlinear probe | | | | |
| DINOv2 | 0.67 | **0.91** | 0.81 | **0.75** |
| SigLIP | **0.69** | 0.78 | **0.84** | 0.73 |
| ConvNeXt | 0.59 | 0.69 | 0.80 | 0.71 |
| CLIP | 0.53 | 0.69 | 0.77 | 0.68 |
| DreamSim | 0.60 | 0.62 | 0.60 | 0.68 |
| DeiT III | 0.50 | 0.71 | 0.72 | 0.68 |
| DINO | 0.63 | 0.67 | 0.70 | 0.66 |
| SAM | 0.52 | 0.62 | 0.71 | 0.59 |
| MASt3R | 0.57 | 0.56 | 0.56 | 0.52 |
| DUSt3R | 0.58 | 0.55 | 0.55 | 0.54 |
| MAE | 0.61 | 0.66 | 0.61 | 0.60 |
| VGGT | 0.52 | 0.54 | 0.53 | 0.50 |
| Mean (nonlinear) | 0.56 | 0.67 | 0.70 | 0.64 |

Table 1: Extended Probing Analysis (Linear vs. Nonlinear) with multi-view pretrained models. We compare performance across standard 2D foundation models and newly added multi-view pretrained models (VGGT, DUSt3R, MASt3R). Featurizers are sorted by the combined rank of accuracies. **Bold** indicates the best result per column/section; underline indicates the second best.

| Method | Plat. | Arch. | Cat. | John. | KP | Stel. | Comp. | NonConv. |
|---|---|---|---|---|---|---|---|---|
| Simple prompt | | | | | | | | |
| LLaVA-3D (Syn) | 20% | 0.0% | 0.0% | 0.0% | 0.0% | 0.0% | 0.0% | 0.0% |
| LLaVA-3D (Wild) | 20% | 0.0% | 0.0% | 0.0% | 0.0% | 0.0% | 0.0% | 0.0% |
| ShapeLLM | 40% | 7.6% | 0.0% | 0.0% | 0.0% | 0.0% | 0.0% | 0.0% |
| PointLLM | 40% | 7.6% | 7.6% | 1.0% | 25% | 0.0% | 0.0% | 3.8% |
| CoT prompt | | | | | | | | |
| LLaVA-3D (Syn) | 0% | 0.0% | 0.0% | 1.0% | 0.0% | 0.0% | 0.0% | 0.0% |
| LLaVA-3D (Wild) | 20% | 7.6% | 0.0% | 0.0% | 0.0% | 0.0% | 0.0% | 0.0% |
| ShapeLLM | 20% | 7.6% | 0.0% | 1.8% | 0.0% | 0.0% | 0.0% | 0.0% |
| PointBind & PointLLM | 0.0% | 15.3% | 0.0% | 0.0% | 0.0% | 0.0% | 0.0% | 1.9% |

Table 2: Zero-Shot Classification with 3D-aware VLMs. We evaluate LLaVA-3D (using synthetic/wild images) and point-cloud-based models (ShapeLLM, PointLLM, utilizing ground truth point clouds). Performance is reported for both Simple and Chain-of-Thought (CoT) prompting strategies.

| Method | Plat. | Arch. | Cat. | John. | KP | Stel. | Comp. | NonConv. |
|---|---|---|---|---|---|---|---|---|
| Synthetic Images | | | | | | | | |
| Baseline Prompt | | | | | | | | |
| GPT-4o (SV) | 80.0% | 7.6% | 0.0% | 0.0% | 50.0% | 0.0% | 10.0% | 0.0% |
| GPT-4o (MV) | 100.0% | 7.6% | 0.0% | 4.0% | 25.0% | 0.0% | 0.0% | 1.9% |
| GPT-5-mini (SV) | 60.0% | 7.6% | 0.0% | 1.4% | 0.0% | 0.0% | 0.0% | 0.0% |
| GPT-5-mini (MV) | 80.0% | 15.3% | 7.6% | 2.1% | 0.0% | 2.9% | 10.0% | 0.0% |
| Chain-of-Thought (CoT) Prompt | | | | | | | | |
| GPT-4o (SV) | 100.0% | 0.0% | 0.0% | 0.0% | 50.0% | 2.9% | 0.0% | 0.0% |
| GPT-4o (MV) | 100.0% | 0.0% | 0.0% | 2.9% | 25.0% | 2.9% | 0.0% | 1.9% |
| GPT-5-mini (SV) | 60.0% | 7.6% | 0.0% | 1.4% | 25.0% | 0.0% | 10.0% | 0.0% |
| GPT-5-mini (MV) | 100.0% | 23.0% | 15.3% | 5.0% | 25.0% | 2.9% | 10.0% | 0.0% |
| Wild Images | | | | | | | | |
| Baseline Prompt | | | | | | | | |
| GPT-4o (SV) | 20.0% | 7.6% | 0.0% | 2.7% | 50.0% | 0.0% | 16.7% | 0.0% |
| GPT-4o (MV) | 40.0% | 0.0% | 0.0% | 3.4% | 25.0% | 0.0% | 16.7% | 1.9% |
| GPT-5-mini (SV) | 20.0% | 23.0% | 7.6% | 0.0% | 0.0% | 0.0% | 0.0% | 0.0% |
| GPT-5-mini (MV) | 20.0% | 23.0% | 7.6% | 3.4% | 25.0% | 0.0% | 16.7% | 0.0% |
| Chain-of-Thought (CoT) Prompt | | | | | | | | |
| GPT-4o (SV) | 20.0% | 7.6% | 0.0% | 0.0% | 50.0% | 0.0% | 0.0% | 0.0% |
| GPT-4o (MV) | 40.0% | 15.3% | 0.0% | 2.7% | 25.0% | 0.0% | 16.7% | 1.9% |
| GPT-5-mini (SV) | 0.0% | 0.0% | 7.6% | 2.0% | 25.0% | 0.0% | 0.0% | 0.0% |
| GPT-5-mini (MV) | 0.0% | 30.7% | 15.3% | 6.9% | 50.0% | 0.0% | 16.7% | 0.0% |

Table 3: Ablation Study on Input Modality and Prompting. We compare zero-shot classification accuracy for ChatGPT-4o and ChatGPT-5-mini using Single-View (SV) vs. Multi-View (MV) inputs, and Baseline vs. Chain-of-Thought (CoT) prompts across synthetic and wild domains.

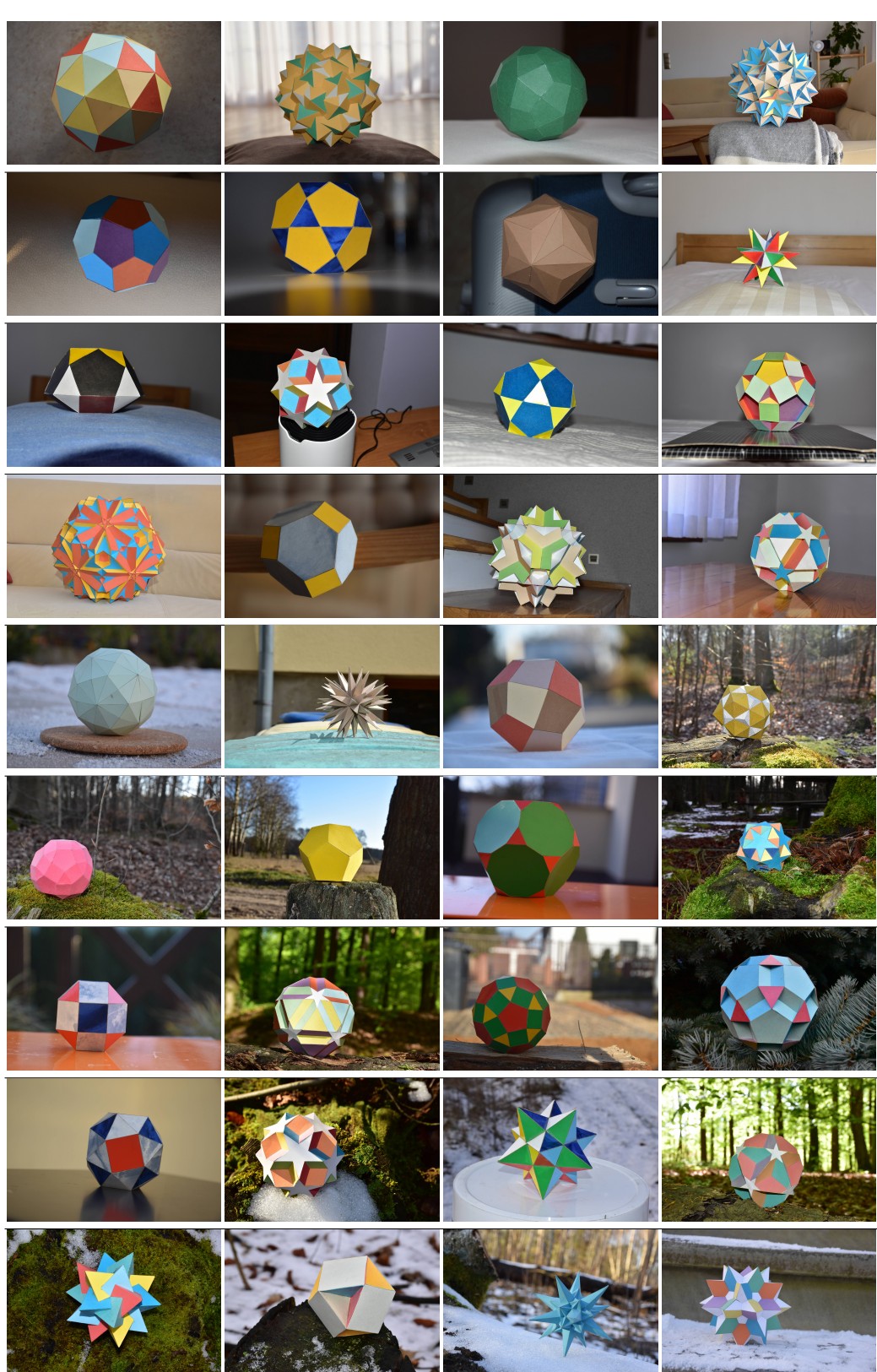

Table 4: **Additional Real-World Examples.** Selected samples from the Wild dataset demonstrating diverse environmental conditions, including indoor, outdoor scenes, and varying lighting.

| Group | # | Key Features | Examples from GIQ |
|---|---|---|---|
| Platonic | 5 | Congruent regular polygonal faces; vertex-transitive, face-transitive |  |
| Archimedean | 13 | Vertex-transitive; faces are regular polygons of different types |  |
| Catalan | 13 | Duals of Archimedean solids; face-transitive, but not vertex-transitive |  |
| Johnson | 92 | Convex polyhedron; regular polygonal faces |  |
| Stellations | 48 | Formed by extending faces or edges; Nonconvex |  |
| Kepler-Poinsot | 4 | Regular star polyhedra; specific stellations of Platonic solids |  |
| Compounds | 10 | Symmetric combination of multiple polyhedra |  |
| Uniform non-convex | 53 | Nonconvex; regular polygonal faces, vertex-transitive |  |

Table 5: Extended summary of polyhedral groups in the GIQ dataset, with counts of 3D shapes, key geometric features, and additional representative samples.

| Symmetry Element | Split | Positives | Negatives | Pos./Neg. Ratio |
|---|---|---|---|---|
| Central point reflection | Train | 1168 | 480 | 2.43 |
| 5-fold rotation | Train | 432 | 1216 | 0.36 |
| 4-fold rotation | Train | 1424 | 224 | 6.36 |
| Central point reflection | Test | 752 | 180 | 4.18 |
| 5-fold rotation | Test | 148 | 784 | 0.19 |
| 4-fold rotation | Test | 716 | 216 | 3.31 |

Table 6: Detailed composition of training and test dataset splits used for 3D symmetry detection experiments (representative statistics from Fold 1 of the 5-fold cross-validation). For each considered symmetry element (central point reflection, 5-fold rotation, and 4-fold rotation), the number of positive and negative samples, as well as the corresponding positive-to-negative ratio, is provided.

| Featurizer | Central point reflection | | 5-fold rotation | | 4-fold rotation | |
|---|---|---|---|---|---|---|
| | Syn | Wild | Syn | Wild | Syn | Wild |
| DINOv2 | 0.85 | 0.73 | 0.97 | **0.85** | 0.96 | **0.93** |
| CLIP | 0.82 | **0.74** | 0.80 | 0.78 | 0.74 | 0.69 |
| ConvNext | 0.76 | 0.62 | 0.93 | **0.85** | 0.90 | 0.76 |
| SigLip | 0.69 | 0.59 | 0.86 | 0.79 | 0.78 | 0.74 |
| MAE | 0.70 | 0.66 | 0.82 | 0.71 | 0.74 | 0.71 |
| DeiT III | 0.75 | 0.65 | 0.82 | 0.72 | 0.70 | 0.65 |
| DreamSim | 0.77 | 0.71 | 0.78 | 0.71 | 0.68 | 0.63 |
| SAM | 0.74 | 0.73 | 0.79 | 0.68 | 0.68 | 0.59 |
| DINO | 0.66 | 0.62 | 0.88 | 0.71 | 0.87 | 0.61 |

Table 7: Extension of main-paper figure: balanced accuracies ($0.5 \cdot \frac{\text{TP}}{P} + 0.5 \cdot \frac{\text{TN}}{N}$) for linear probes trained on synthetic images and evaluated on both synthetic (Syn) and real-world (Wild) images for central point reflection, 5-fold rotation, and 4-fold rotation. Including Syn results quantifies the Syn→Wild generalization gap, which varies by featurizer and symmetry. Featurizers are sorted by the combined rank of Wild-set accuracies across the three symmetry types. **Bold** indicates best accuracy; underline indicates second best.

| Input | Frontier Model Reasoning |
|---|---|
| 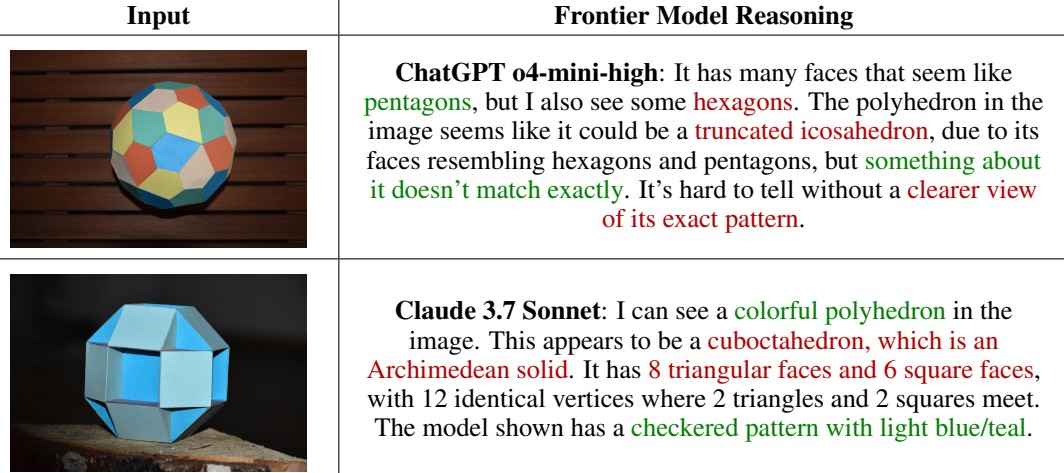 | **ChatGPT o4-mini-high**: It has many faces that seem like pentagons, but I also see some hexagons. The polyhedron in the image seems like it could be a truncated icosahedron, due to its faces resembling hexagons and pentagons, but something about it doesn't match exactly. It's hard to tell without a clearer view of its exact pattern. |
| | **Claude 3.7 Sonnet**: I can see a colorful polyhedron in the image. This appears to be a cuboctahedron, which is an Archimedean solid. It has 8 triangular faces and 6 square faces, with 12 identical vertices where 2 triangles and 2 squares meet. The model shown has a checkered pattern with light blue/teal. |

Figure 1: Examples of failure cases illustrating errors in reasoning by frontier models that led to misclassification of polyhedra. These cases highlight systematic mistakes in geometric recognition, such as misidentification of face geometry, convexity, and compound structures. Text highlighted in green indicates correct statements, while text in red indicates incorrect reasoning.

Table 8: Additional qualitative results for monocular 3D reconstruction, supplementing evaluations presented in the main paper. Columns depict, from left to right: the input 2D image, followed by front-view and side-view renderings of reconstructions from Shap-E Jun & Nichol (2023), Stable Fast 3D Boss et al. (2024), and OpenLRM He & Wang (2023) methods. Each pair of rows shows synthetic (top) and real-world (bottom) images of selected polyhedra: truncated icosidodecahedron (Archimedean solid), compound of five tetrahedra (compound), compound of five octahedra (compound), great triambic icosahedron (stellation), and final stellation of the icosahedron (stellation).

| Category | Metric | Synthetic | | | Wild | | |
|---|---|---|---|---|---|---|---|
| | | Shap-E | SF3D | OpenLRM | Shap-E | SF3D | OpenLRM |
| Platonic | F-score ↑ | 0.367 | 0.521 | 0.626 | 0.380 | 0.378 | 0.224 |
| | Hausdorff Distance ↓ | 0.083 | 0.047 | 0.043 | 0.158 | 0.123 | 0.173 |
| | Chamfer Distance ↓ | 0.001 | 0.001 | 0.001 | 0.006 | 0.004 | 0.008 |
| Archimedean | F-score ↑ | 0.355 | 0.477 | 0.586 | 0.309 | 0.348 | 0.175 |
| | Hausdorff Distance ↓ | 0.083 | 0.052 | 0.062 | 0.134 | 0.137 | 0.215 |
| | Chamfer Distance ↓ | 0.001 | 0.002 | 0.001 | 0.008 | 0.007 | 0.009 |
| Catalan | F-score ↑ | 0.361 | 0.478 | 0.597 | 0.257 | 0.365 | 0.172 |
| | Hausdorff Distance ↓ | 0.086 | 0.059 | 0.050 | 0.156 | 0.120 | 0.184 |
| | Chamfer Distance ↓ | 0.001 | 0.002 | 0.002 | 0.009 | 0.005 | 0.007 |
| Stellations | F-score ↑ | 0.259 | 0.231 | 0.239 | 0.162 | 0.096 | 0.191 |
| | Hausdorff Distance ↓ | 0.119 | 0.179 | 0.158 | 0.313 | 0.292 | 0.297 |
| | Chamfer Distance ↓ | 0.002 | 0.011 | 0.007 | 0.051 | 0.020 | 0.009 |
| Kepler-Poinsot | F-score ↑ | 0.255 | 0.292 | 0.258 | 0.257 | 0.124 | 0.245 |
| | Hausdorff Distance ↓ | 0.115 | 0.116 | 0.120 | 0.147 | 0.254 | 0.218 |
| | Chamfer Distance ↓ | 0.002 | 0.001 | 0.002 | 0.002 | 0.017 | 0.006 |
| Compounds | F-score ↑ | 0.272 | 0.253 | 0.252 | 0.220 | 0.113 | 0.184 |
| | Hausdorff Distance ↓ | 0.110 | 0.136 | 0.135 | 0.172 | 0.294 | 0.271 |
| | Chamfer Distance ↓ | 0.001 | 0.002 | 0.002 | 0.003 | 0.023 | 0.013 |
| Uniform Nonconvex | F-score ↑ | 0.263 | 0.194 | 0.250 | 0.232 | 0.120 | 0.182 |
| | Hausdorff Distance ↓ | 0.122 | 0.144 | 0.119 | 0.145 | 0.257 | 0.186 |
| | Chamfer Distance ↓ | 0.002 | 0.005 | 0.002 | 0.004 | 0.021 | 0.008 |

Table 9: Quantitative comparison of state-of-the-art monocular 3D reconstruction methods (Shap-E Jun & Nichol (2023), Stable Fast 3D Boss et al. (2024) (SF3D), OpenLRM He & Wang (2023)) evaluated on synthetic and real-world ("wild") datasets across various polyhedral categories. Consistently low F-scores (below 0.6) indicate substantial reconstruction difficulties, emphasizing the complexity of accurately capturing detailed geometric structures in polyhedra and highlighting areas needing significant methodological improvements.

| Featurizer | Concat $(e_1 \| e_2)$ | Subtraction $(e_1 - e_2)$ | Absolute $(|e_1 - e_2|)$ |
|:---:|:---:|:---:|:---:|
| CLIP | 0.5065 | 0.5786 | 0.9400 |
| ConvNext | 0.5188 | 0.5468 | 0.9306 |
| DeiT III | 0.4804 | 0.5537 | 0.9473 |
| DINO | 0.4616 | 0.5380 | 0.9674 |
| DINOv2 | 0.4991 | 0.5693 | 0.9706 |
| DreamSim | 0.5221 | 0.5983 | 0.9843 |
| MAE | 0.5240 | 0.5247 | 0.9768 |
| SAM | 0.5035 | 0.5516 | 0.9519 |
| SigLip | 0.5399 | 0.5615 | 0.9594 |

Table 10: Balanced accuracy of linear probing approaches on pairwise embeddings for the Mental Rotation Test (MRT). Given two embeddings $e_1, e_2$ from each featurizer, we form input features by concatenation ($e_1 \| e_2$), subtraction ($e_1 - e_2$), or absolute difference ($|e_1 - e_2|$), followed by a linear classifier. Results are reported on a simplified "trivial" setting, with synthetic-only image pairs and shapes randomly split into 80% train and 20% test.

| Featurizer | Train (syn,syn)→Test (syn,syn) | Train (syn,syn)→Test (syn,wild) | Train (syn,wild)→Test (syn,wild) |
|---|---|---|---|
| **Absolute Difference** $\lvert e_1 - e_2 \rvert$ | | | |
| CLIP | 0.64 | 0.44 | 0.55 |
| ConvNext | 0.70 | 0.47 | 0.65 |
| DeiT III | 0.70 | 0.48 | 0.56 |
| DINO | 0.74 | 0.44 | 0.58 |
| DINOv2 | 0.79 | 0.44 | 0.60 |
| DreamSim | 0.77 | 0.44 | 0.62 |
| MAE | 0.76 | 0.44 | 0.58 |
| SAM | 0.77 | 0.44 | 0.57 |
| SigLip | 0.76 | 0.44 | 0.64 |
| **Subtraction** $(e_1 - e_2)$ | | | |
| CLIP | 0.36 | 0.39 | 0.56 |
| ConvNext | 0.51 | 0.49 | 0.54 |
| DeiT III | 0.45 | 0.46 | 0.53 |
| DINO | 0.48 | 0.49 | 0.49 |
| DINOv2 | 0.53 | 0.51 | 0.45 |
| DreamSim | 0.38 | 0.34 | 0.55 |
| MAE | 0.37 | 0.34 | 0.53 |
| SAM | 0.47 | 0.48 | 0.50 |
| SigLip | 0.50 | 0.44 | 0.54 |
| **Concatenation** $(e_1 \Vert e_2)$ | | | |
| CLIP | 0.35 | 0.56 | 0.58 |
| ConvNext | 0.46 | 0.50 | 0.52 |
| DeiT III | 0.41 | 0.54 | 0.54 |
| DINO | 0.52 | 0.52 | 0.54 |
| DINOv2 | 0.57 | 0.48 | 0.60 |
| DreamSim | 0.38 | 0.55 | 0.55 |
| MAE | 0.41 | 0.52 | 0.53 |
| SAM | 0.37 | 0.56 | 0.59 |
| SigLip | 0.36 | 0.44 | 0.54 |

Table 11: Accuracy on the Mental Rotation Test (MRT) evaluated on the *hard* test set, where only pairs of visually similar shapes are considered. "Train (X)→Test (Y)" denotes the training and testing domains. Results are presented for absolute difference ($\lvert e_1 - e_2 \rvert$), raw subtraction ($e_1 - e_2$), and feature concatenation ($e_1 \Vert e_2$). Absolute difference performed best overall.

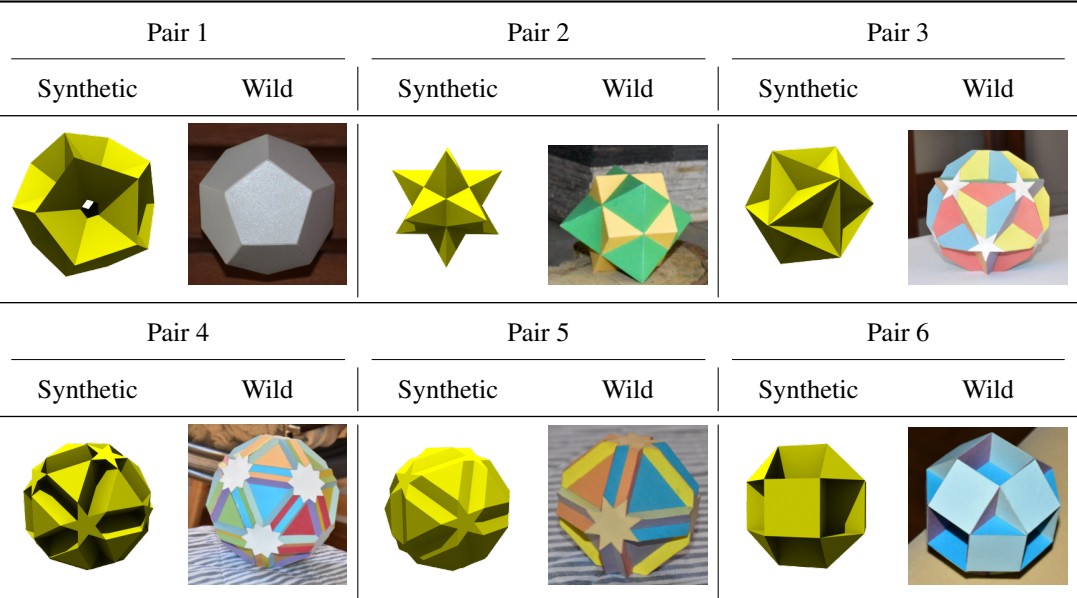

| Pair 1 | | Pair 2 | | Pair 3 | |
|---|---|---|---|---|---|
| Synthetic | Wild | Synthetic | Wild | Synthetic | Wild |

| Pair 4 | | Pair 5 | | Pair 6 | |
|---|---|---|---|---|---|
| Synthetic | Wild | Synthetic | Wild | Synthetic | Wild |

Table 12: Samples of visually and geometrically similar synthetic-wild shape pairs used in the *hard* test set. Pairs were manually selected based on structural and visual similarities, such as shared symmetries, vertex configurations, and derivation from common polyhedra (e.g., pair 6: Small cubicuboctahedron and Small rhombihexahedron, both derived from the rhombicuboctahedron).

| Category | Gemini 2.5 Pro | | ChatGPT o4-mini-high | | ChatGPT o3 | | Claude 3.7 Sonnet | |
|---|---|---|---|---|---|---|---|---|
| | Syn | Wild | Syn | Wild | Syn | Wild | Syn | Wild |
| Platonic | 0.60 | 0.80 | 0.60 | 0.60 | **1.00** | **1.00** | 0.60 | 0.60 |
| Archimedean | 0.53 | **0.61** | 0.23 | 0.31 | **0.61** | 0.54 | 0.21 | 0.23 |
| Catalan | **0.15** | **0.15** | **0.15** | **0.15** | 0.08 | 0.08 | 0.09 | 0.08 |
| Johnson | 0.20 | **0.18** | 0.11 | 0.12 | **0.21** | **0.18** | 0.11 | 0.11 |
| Kepler-Poinsot | **1.00** | **1.00** | 0.25 | 0.25 | 0.25 | 0.25 | 0.25 | 0.25 |
| Stellations | **0.41** | **0.43** | 0.05 | 0.07 | 0.23 | 0.21 | 0.27 | 0.26 |
| Compounds | 0.16 | 0.17 | 0.00 | 0.00 | 0.16 | **0.33** | **0.33** | **0.33** |
| Uniform non-convex | **0.08** | **0.09** | 0.00 | 0.00 | 0.05 | 0.06 | 0.00 | 0.00 |

Table 13: Accuracy (%) of various frontier models on 0-shot classification across polyhedron categories, reported on synthetic (Syn) and in-the-wild (Wild) images. The Syn→Wild gap is generally small across categories and models, indicating comparable performance across domains.