# OpenReview forum: "GIQ: Benchmarking 3D Geometric Reasoning of Vision Foundation Models with Simulated and Real Polyhedra"
_ICLR.cc/2026/Conference — ICLR 2026 Poster_

### Official Review · Reviewer_MgEn · 2025-10-17

**Soundness:** 3
**Presentation:** 3
**Contribution:** 3
**Rating:** 6
**Confidence:** 4

**Summary:**

This paper proposed a new benchmark, the Geometric IQ Test (GIQ), with simulated and physical polyhedra to evaluate the geometric reasoning capabilities of vision and vision-language models on four tasks. In addition, this work identifies limitations of existing models through an empirical study on the novel benchmark.

**Strengths:**

- The GIQ dataset consists of synthetic and real-world images with corresponding 3D meshes that cover 224 unique polyhedra (with various levels of complexity and shapes). It is designed to evaluate visual reasoning with respect to 3D object geometry properties. The dataset of polyhedra provides unambiguous ground truth for evaluation.
- This work provides extensive empirical evidence on existing models, demonstrating the current limitations of both vision and vision-language models. The experiment covers four tasks: (1) monocular 3D reconstruction, 3D symmetry detection, mental rotation tests, and zero-shot polyhedron classification.
- The new polyhedra-based benchmark enables the research communities to systematically evaluate vision model performance on fundamental and fine-grained geometric reasoning (e.g., symmetry, complexity, and geometric properties).

**Weaknesses:**

- The test split exclusively contains views from 26 unique polyhedral shapes not present in the training set. How are these 26 shapes selected?  Would it be fairer to conduct n-fold validation (with different test set on n runs) to avoid potential biased findings due to the dataset split?
- It is not entirely clear to me how the new benchmark finding informs the model's performance for arbitrary 3D shapes for real-world applications. I understand that using a polyhedral shape provides highly controllable samples and reliable ground truth. It would be great if the author could provide a discussion on how the findings translate to real-world scenarios for arbitrary objects that do not satisfy the polyhedral shapes' properties.
- The inclusion of real-world images taken with paper models is a good initiative. However, from the provided images, the surface is based on non-reflective material. Hence, it seems to be similar to synthetic data but with some variance in shadow conditions. In addition, wild images were preprocessed via centre cropping and background removal. In such a scenario, it is uncertain how "difficult" the wild image is compared to synthetic data. It would be useful to analyse the results with the synthetic and wild images separately. In a more challenging and ideal case, I wish the benchmark could have real-world images taken with polyhedral models built with reflective material.

**Questions:**

- For the monocular 3D reconstruction task, each of the models was trained on millions of 3D assets. However, I believe this model is not aware of polyhedral shapes nor trained with polyhedral shapes. Hence, these models might be "overthinking" about the underlying 3D shape of the provided image, and not taking advantage of polyhedral space properties in 3D model generation. Instead of just describing the observation from the generated results, I would like the author to discuss the reasons why these models are performing badly, and what insights to inform future research. In addition, what is the data split for the monocular 3D reconstruction task?
- Figure 2 and Table 1 (appendix) show a total of 238 samples from 8 groups. This number is more than the 224 unique polyhedra shapes stated in the paper. Please clarify.
- It would be helpful to provide a background or reference for 4-fold rotational symmetry and 5-fold rotational symmetry. This will help the reader to understand the task better.

---

> ### Author Response · Authors · 2025-11-22
> **Response to Reviewer MgEn**
>
> We thank the reviewer for the positive assessment and for recognizing that GIQ enables systematic evaluation of vision model performance. We appreciate the constructive feedback regarding dataset splits and real-world applicability.
>
> **1\. Test Split and N-Fold Validation**
>
> We initially selected the 26 test shapes to strictly balance the ratio of positive and negative examples for specific symmetry attributes (see Table 2 in Supplementary Material). However, we agree that n-fold validation provides a more robust estimate.
>
> Following the reviewer’s advice, we performed 5-fold cross-validation for the symmetry detection task. The results confirm our original findings and are detailed in the General Response (to all reviewers).
>
> **2\. Generalization to Arbitrary Real-World Shapes**
>
> We thank the reviewer for this insightful question. We view GIQ as a "geometric litmus test."
>
> While there is a gap between "primitives" and arbitrary organic shapes, geometric primitives are the building blocks of complex reasoning. If a model cannot recover the planar faces of a perfect cube or the symmetry of a dodecahedron, its "3D understanding" is likely a fragile approximation based on texture rather than true spatial intelligence.
>
> Real-world objects are governed by these same principles. Distinguishing convexity (e.g., a bowl vs. a ball) or identifying symmetry (e.g., functional parts of a machine) relies on the same core "geometric arithmetic" tested in GIQ. Failure here implies the model is not ready for the "geometric calculus" of complex scenes. We will add this discussion to the final paper. We acknowledge, however, that establishing a precise quantitative correlation between performance on GIQ primitives and specific downstream tasks involving arbitrary organic shapes is complex. While we posit that GIQ measures necessary foundational skills, future work is needed to empirically map these metrics to broader real-world performance benchmarks. We will add this discussion to the final paper.
>
> **3\. Wild Images: Difficulty and Preprocessing**
>
> We clarify a key misunderstanding: preprocessing (background removal/cropping) was applied only for the Monocular 3D Reconstruction task to ensure a fair comparison with SOTA baselines which expect object-centered inputs with no background. For all other tasks (Symmetry, Mental Rotation, Classification), the "Wild" images retained their original complex backgrounds. These include diverse indoor and outdoor scenes, and varying weather (snow, direct sun). While the paper models are matte (to test geometry, not specularity), they were often photographed on reflective surfaces, introducing complex lighting interactions absent in synthetic data. We will include additional examples in the supplementary material to better illustrate this diversity.
>
> We agree that separate analysis for synthetic vs wild is vital. Tables 3, 9, 11, and 13 in the Supplementary Material provide explicit breakdowns of Synthetic vs. Wild performance.
>
> **4\. Monocular 3D Reconstruction Failure Modes**
>
> While Objaverse (training data for baselines) contains some polyhedra (Platonic, Archimedean and other solids are available in Sketchfab which is part of Objaverse), it is dominated by noisy real-world scans. Consequently, models have learned a prior for "imperfect, noisy surfaces" rather than "mathematical exactness." They fail to apply constraints like planarity or strict symmetry because they are optimizing for a visual average rather than geometric structure. This suggests that scale alone is insufficient. Future research requires hybrid training involving mathematically generated data to force the learning of geometric constraints (planarity, sharp edges).
>
> Monocular 3D reconstruction was a zero-shot evaluation using pre-trained public models; GIQ served purely as an out-of-distribution test set.
>
> **5\. Shape Count (238 vs 224)**
>
> The dataset contains 224 unique geometric shapes. The sum in the table (238) is higher because categorization is not mutually exclusive; for example, Kepler-Poinsot solids are a subset of stellations. We will clarify this overlap in the caption.
>
> **6\. Symmetry Definitions**
>
> We will expand the manuscript to explicitly define n-fold rotational symmetry as the property where an object maps onto itself after a rotation of 360/n.

---

> > ### Comment · Reviewer_MgEn · 2025-11-26
> > **Acknowledgement of rebuttal**
> >
> > I sincerely thank you for the authors' response. The response addressed the concerns I had in the initial review.
> >
> > I will carefully consider the response, the submitted manuscript, and the fellow reviewers' assessments in making the final decision.

---

### Official Review · Reviewer_GJKD · 2025-10-31

**Soundness:** 3
**Presentation:** 3
**Contribution:** 3
**Rating:** 6
**Confidence:** 2

**Summary:**

The paper introduces the GIQ (Geometric IQ Test) benchmark. This tool specifically evaluates the 3D geometric reasoning abilities of modern vision foundation models (VLMs). The GIQ dataset contains images and 3D meshes for 224 unique polyhedra. These shapes range systematically in complexity and symmetry. They cover simple Platonic solids up to complex structures like stellations. Both synthetic renderings and real-world photographs are included. The authors conducted four key experiments on state-of-the-art models. The results showed a significant gap in performance. Models struggle with explicit, robust geometric tasks despite high scores on common benchmarks. The paper concludes that GIQ is a crucial diagnostic tool. It is designed to guide the development of future, more geometry-aware foundation models.

**Strengths:**

1. It provides the critical insight that a model's implicit ability to encode a feature (demonstrated by successful linear probing for symmetry) does not translate into explicit, robust geometric reasoning in other tasks. The results are quite interesting.
2. For GIQ, it uses polyhedra with well-defined properties (symmetry groups, face types) to provide precise, unambiguous ground truth for fine-grained geometric evaluation, which is lacking in large, existing 3D datasets.The constructed dataset is diverse, including both controlled synthetic renderings (Mitsuba PBR) and "wild images" of physical paper models captured in diverse real-world conditions. I appreciate the contribution of this proposed benchmark and its comprehensive dataset.

**Weaknesses:**

1. One concern is the potential VLM prompt ambiguity. The exact zero-shot prompt methodology for testing VLMs is not detailed in the provided text. The reported low accuracy could potentially be influenced by sub-optimal or ambiguous prompting rather than a pure geometric failure of the models.
2. While polyhedra are rigorous, their highly regular and stylized nature may not fully capture the complexity and irregularity of general, arbitrary objects found in the real world. One potential concern is how to better represent the 3D objects in any format, such as liquid. This may remain as the future work.

**Questions:**

See weakness

---

> ### Author Response · Authors · 2025-11-22
> **Response to Reviewer GJKD**
>
> We thank the reviewer for the positive assessment and for appreciating the recognition of our dataset's diversity and the value of rigorous ground truth. Below, we address the reviewer's specific concerns.
>
> **1. VLM Prompt Ambiguity**
>
> We thank the reviewer for this suggestion. To ensure that the reported low accuracy reflects a lack of geometric reasoning rather than sub-optimal prompting, we conducted additional experiments comparing our original baseline against a detailed Chain-of-Thought (CoT) prompt.
>
> Baseline Prompt: *"What is the name of this polyhedron?"*
>
> Chain-of-Thought Prompt: *"Let's identify this polyhedron by thinking step-by-step: Convexity: First, analyze its overall shape. Is this a convex polyhedron or is it non-convex? Symmetry and Rotation: Second, describe its symmetries. What rotational symmetries does it have? Does it have planes of reflectional symmetry? Faces and Vertices: Third, describe its components. What is the shape of each face, and how many faces meet at each vertex? Conclusion: Based on this analysis of its convexity, symmetry, and components, what is the precise name of this polyhedron?"*
>
> As detailed in the results table in our General Response, the explicit CoT prompt did not significantly improve performance. In many cases, models hallucinated the intermediate steps (e.g., describing non-existent hexagonal faces) to justify an incorrect conclusion. This confirms that the failure stems from a fundamental inability to ground geometric concepts in visual data, rather than ambiguous instructions.
>
> **2. Polyhedra vs. Arbitrary Real-World Objects**
>
> We thank the reviewer for this thoughtful comment. We agree that polyhedra are a specific, highly structured class of objects that do not capture the full irregularity of arbitrary shapes or non-rigid forms like liquids.
>
> However, this choice was deliberate. Our goal with GIQ was to create a diagnostic benchmark for fundamental geometric reasoning. Polyhedra are uniquely suited for this because they provide unambiguous, mathematically precise ground truth.
>
> If a model cannot correctly reason about the fundamental properties of a "simple" shape like a cube—as our reconstruction results show—it cannot be expected to truly understand the complex, fluid geometry of an arbitrary or liquid object.
>
> We agree that extending evaluations to non-rigid objects is an exciting avenue for future research, and we will add a discussion point to the paper clarifying the scope of our benchmark and motivating this direction.

---

### Official Review · Reviewer_pamt · 2025-10-31

**Soundness:** 4
**Presentation:** 4
**Contribution:** 4
**Rating:** 8
**Confidence:** 4

**Summary:**

This paper presents a new benchmark for 3D geometric reasoning in modern vision and vision-language models. This benchmark includes a key dataset of 224 polyhedra spanning Platonic, Archimedean, Johnson, Catalan, Kepler-Poinsot, stellated, and compound solids. They are represented in both synthetic renderings (via Mitsuba) and real-world photographs (of physical paper models). Many tasks are included in a comprehensive evaluation, inc., monocular 3D reconstruction, 3D symmetry detection, mental rotation and zero-shot classification.

The most valuable and interesting part of this paper is the conclusion. Results reveal that while pretrained encoders sometimes implicitly capture symmetry, most models fail catastrophically on real 3D reasoning tasks, especially reconstruction, rotation, and geometric classification, even on simple solids.

**Strengths:**

I believe this is an interesting and significantly meaningful paper.
It sets a geometrical IQ (G-IQ) test for modern vision models and vlms in 3D reconstruction. GIQ fills a clear gap: existing 3D datasets (Objaverse, OmniObject3D, GSO) test recognition and reconstruction, but not reasoning about geometry or symmetry. Polyhedra are a brilliant choice since they offer mathematically clean ground truth, structured complexity, and interpretability, and the dataset design (Mitsuba renders + paper models) ensures good control and realism, and they find that DINOv2 captures symmetry implicitly while failing in higher-level reasoning, which is interesting and insightful.

The experiments are comprehensive and rigorous. The paper presentation is clear and easy to follow.

**Weaknesses:**

Since the paper’s premise involves geometric reasoning, a simple human baseline (even small-scale) on the same tasks would help contextualise the human-level intelligence in G-IQ test, which will help to understand how far the modern vision models are away from human-level performance.

For the zero-shot classification, it’s unclear how prompts and outputs were standardised. Did all models get the same prompt verbatim? Were answers normalised (e.g., “cube” vs. “hexahedron”)?

**Questions:**

Could GIQ be used for fine-tuning or instruction-tuning VLMs to improve their geometric intelligence?

---

> ### Author Response · Authors · 2025-11-22
> **Response to Reviewer pamt**
>
> We thank the reviewer for the high rating and for recognizing GIQ as a "significantly meaningful paper" that fills a clear gap in evaluating 3D geometric reasoning. We appreciate the reviewer's observation that our choice of polyhedra offers mathematically clean ground truth and structured complexity. Below, we address the reviewer’s specific comments.
>
> **1. Human Baseline for Mental Rotation**
>
> We agree that a human baseline provides essential context. To address this, we conducted a study with 42 participants.
> Protocol: We developed a web interface presenting the exact same image pairs used in the paper. Participants answered 25 questions (5 from the "easy" set, 20 from the "hard" set) with a 30-second time limit per question. Options were "Same", "Different", or "I don't know" (which was scored as 0).
>
> Results:
>
> Easy Set: Mean accuracy was 97.56%, confirming participants understood the task.
>
> Hard Set: Mean accuracy was 68.05% (Std Dev: 0.11), with top performers scoring 18/20.
>
> Comparison:
>
> While the best non-linear probe (SigLIP) performs on par with the average human (~69%), it is worth noting that 68% of human participants scored higher than the best performing foundation model. This confirms that while models are closing the gap on average,  top-performing human participants still outperform them on complex instances. We will include these results in the final paper.
>
> **2. Zero-Shot Standardization**
>
> We clarify that standardization was strictly enforced:
>
> Prompts: All models received the verbatim prompt: "What is the name of this polyhedron?"
>
> Normalization: We accounted for synonyms (e.g., Cube/Hexahedron, Stella Octangula/Stellated Octahedron). We manually verified that no models produced correct answers using synonyms outside our dictionary.
>
> Additionally, following suggestions from other reviewers, we tested (a) Chain-of-Thought prompts explicitly requesting reasoning on convexity, symmetry, and face counts, and (b) Multi-view inputs. As detailed in the General Response, explicit reasoning prompts yielded minimal gains. Multi-view inputs yielded helpful but limited gains for low-symmetry shapes (specifically: Johnson solids) by resolving some ambiguities but did not significantly improve performance on high-symmetry solids (Platonic/Archimedean), where a single view suffices for identification.
>
> **3. Using GIQ for Fine-tuning/Instruction-tuning**
>
> We believe the methodology behind GIQ is well-suited for instruction-tuning, though the current evaluation set has limitations for this purpose.
>
> The dataset's structured ontology allows for the auto-generation of precise Q&A pairs and Chain-of-Thought (CoT) training data, which could ground models in explicit geometric properties.
>
> However, as the reviewer implies, the current dataset size (224 shapes) is optimized for evaluation rather than training. Using GIQ directly for fine-tuning risks overfitting and memorization. Therefore, while GIQ establishes the necessary ontology and data generation pipeline, a training-focused version would require scaling up to a broader distribution of "wild" appearances to ensure robust generalization. We view this as a promising direction for future work.
>
> We thank the reviewer again for the constructive feedback and the encouraging assessment of our work.

---

> > ### Comment · Reviewer_pamt · 2025-11-26
> > **Thanks for the rebuttal**
> >
> > I believe the authors addressed my concerns and I am inclined to accptance.

---

### Official Review · Reviewer_15Vf · 2025-11-02

**Soundness:** 3
**Presentation:** 3
**Contribution:** 3
**Rating:** 4
**Confidence:** 3

**Summary:**

The paper proposes a geometric IQ (GIQ) benchmark designed to evaluate the 3D geometric reasoning ability of vision and vision-language foundation models. The dataset includes synthetic and real-world images of 224 unique polyhedra, covering convex, nonconvex, stellated, and compound classes, each captured from multiple viewpoints.
The benchmark defines four diagnostic tasks – monocular 3D reconstruction, symmetry detection, mental rotation, and zero-shot shape classification – to probe different aspects of geometric understanding. A linear probing framework is used to assess how well pretrained visual representations encode geometric properties such as symmetry and shape equivalence.
Experimental results show that while models like DINOv2, CLIP, and GPT-4o exhibit some pattern sensitivity, none demonstrate consistent 3D awareness or symmetry invariance, especially when generalizing from synthetic to real conditions. Overall, the benchmark highlights a significant gap between current foundation models and true geometry-aware visual reasoning.

**Strengths:**

1.The paper is well-motivated and addresses an important gap. Foundation models based on language and/or 2D images are unlikely to develop 3D understanding. The proposed benchmark demonstrates these limitations through controlled experiments.
2.The combination of synthetic and real images provide a well-controlled domain-shift test for evaluating geometric invariance versus appearance sensitivity.

**Weaknesses:**

1. The benchmark primarily evaluates 2D-pretrained encoders (e.g., CLIP, DINOv2, MAE, ConvNeXt) and 2D VLMs (e.g., GPT-4o, Claude, Gemini, Llava), which were never designed to represent 3D structure—so their poor performance is somewhat expected. The study highlights a limitation of 2D pretraining rather than establishing a hierarchy of geometry-aware capabilities.
2.Including geometry-native or 3D-aware models -- such as multi-view pretrained networks (VGGT, DUSt3R, MASt3R, etc.) or 3D-VLMs (PointLLM, SceneLLM, LLaVA-3D) -- would strengthen the conclusions and demonstrate that the benchmark can distinguish true geometric understanding from 2D appearance bias.
3. While the dataset includes both synthetic and real images, evaluations are limited to single-view inputs, which can be ambiguous for many polyhedra. Additionally, there is no exploration of prompted geometric reasoning (e.g., symmetry-aware or chain-of-thought prompts) for VLMs, which might reveal latent reasoning capacity.
4. Linear probing only captures linearly separable features and may not fully reflect a model’s potential for nonlinear geometric reasoning or spatial representation. Complementary analyses (e.g., fine-tuning or nonlinear probes) would make the conclusions stronger.

**Questions:**

1. How do 3D-aware VLMs perform on the benchmark?
2. Does prompting vision-language models to explicitly reason about geometry (e.g., symmetry, rotation, convexity, or step-by-step reasoning) change their performance?
3. How do the models perform with multiview images?

---

> ### Author Response · Authors · 2025-11-22
> **Response to Reviewer 15Vf**
>
> We thank the reviewer for the constructive feedback and for recognizing that our paper is "well-motivated" and effectively uses domain shifts to test geometric invariance. We appreciate the suggestion to broaden our evaluation to geometry-native models, which has significantly strengthened our analysis. Below, we address the reviewer’s specific concerns.
>
> **1. Performance of 3D-Native and Multi-View Models**
>
> We agree that evaluating only 2D-pretrained models might obscure whether failures are due to modality or reasoning deficits. To address this, we expanded our benchmark to include:
>
> Multi-view pretrained networks: We evaluated VGGT, DUSt3R, and MASt3R.
>
> 3D-VLMs: We evaluated LLaVA-3D (image inputs), as well as ShapeLLM and PointBind&PointLLM (using ground truth point clouds as input).
>
> As detailed in the General Response, these geometry-native models did not significantly outperform generalist frontier models on our tasks.
>
> This finding suggests that the "geometric gap" we identified is not merely a limitation of 2D pretraining. Even models with explicit 3D architectural priors struggle with the abstract geometric reasoning (e.g., classifying specific polyhedral classes or identifying symmetries) required by GIQ. This reinforces GIQ’s value as a diagnostic tool for high-level geometric intelligence, distinct from low-level structural reconstruction.
>
> **2. Prompted Geometric Reasoning (CoT)**
>
> We thank the reviewer for this suggestion. To ensure that reported low accuracies reflect a lack of reasoning rather than sub-optimal prompting, we compared our baseline against a detailed Chain-of-Thought (CoT) prompt.
>
> Baseline Prompt: *"What is the name of this polyhedron?"*
>
> Chain-of-Thought Prompt: *"Let's identify this polyhedron by thinking step-by-step: Convexity: First, analyze its overall shape. Is this a convex polyhedron or is it non-convex? Symmetry and Rotation: Second, describe its symmetries. What rotational symmetries does it have? Does it have planes of reflectional symmetry? Faces and Vertices: Third, describe its components. What is the shape of each face, and how many faces meet at each vertex? Conclusion: Based on this analysis of its convexity, symmetry, and components, what is the precise name of this polyhedron?"*
>
> As detailed in the General Response, the explicit CoT prompt did not significantly improve performance. In many cases, models hallucinated intermediate steps (e.g., describing non-existent hexagonal faces) to justify an incorrect conclusion. This confirms that the failure stems from a fundamental inability to ground geometric concepts in visual data, rather than ambiguous instructions.
>
> **3. Non-Linear Probing**
>
> We followed the reviewer's advice and implemented non-linear probes to complement our linear analysis.
>
> Symmetry Detection: Non-linear probes did not yield significant improvements. Average accuracy remained comparable to linear baselines (e.g., Mean CPF: $0.62 \to 0.64$; Mean 5-fold: $0.75 \to 0.70$).
>
> Mental Rotation (MRT): While the average accuracy across featurizers remained near chance (~55.6%), the best-performing model (SigLIP with a non-linear probe) achieved 69% accuracy, performing on par with the average human participant (68%) in our newly added human study. However, it is worth noting that 68% of individual human participants still scored higher than this best-performing model.
>
> **4. Multi-View Performance and Ambiguity**
>
> We conducted additional experiments providing frontier VLMs with multi-view inputs (3 images).
>
> Performance improved for Johnson solids, but we observed no significant gains for other classes (Platonic, Archimedean, Compounds).
>
> The sufficiency of a single view is highly dependent on symmetry:
>
> High-Symmetry Polyhedra: For Platonic, Archimedean, Catalan and Compound solids (possessing high-order symmetries like $I_h$, $O_h$), a single canonical view is informationally complete; the entire structure is "implied" from any non-degenerate angle.
>
> Low-Symmetry Solids: Johnson solids often have low-order symmetries (e.g., $C_{4v}$). For example, a square pyramid viewed from the base is indistinguishable from a square. It was precisely due to this inherent ambiguity that we explicitly excluded Johnson solids from our Monocular Reconstruction and Symmetry Detection experiments in the original submission.
>
> We believe these additional evaluations on 3D-native models, non-linear probes, and multi-view inputs robustly address the reviewer's concerns and further validate the utility of GIQ.

---

### Author Response · Authors · 2025-11-22
**Additional experiments**

We thank all reviewers for their constructive feedback. To address the common questions raised regarding human performance limits, the efficacy of 2D vs. 3D architectures, and the impact of prompting strategies, we have conducted a comprehensive suite of additional experiments detailed below.

**1. Extended Probing Analysis (Linear vs. Non-Linear)**

We expanded our probing analysis to include non-linear probes and added three multi-view pretrained models (VGGT, DUSt3R, MASt3R) to test if geometry-native encoders offer better embeddings for symmetry and rotation tasks.

| Featurizer | Syn-Wild MRT | 4-fold rot | 5-fold rot | CPF |
| :--- | :---: | :---: | :---: | :---: |
| **LINEAR PROBE** | | | | |
| clip | 0.39 | 0.74 | 0.78 | 0.70 |
| convnext | 0.60 | 0.80 | 0.86 | 0.64 |
| deit iii | 0.41 | 0.69 | 0.74 | 0.71 |
| dino | 0.54 | 0.71 | 0.81 | 0.56 |
| dinov2 | 0.43 | 0.93 | 0.84 | 0.80 |
| dreamsim | 0.49 | 0.74 | 0.78 | 0.65 |
| dust3r | 0.53 | 0.63 | 0.67 | 0.49 |
| mae | 0.42 | 0.62 | 0.63 | 0.53 |
| mast3r | 0.48 | 0.64 | 0.70 | 0.58 |
| sam | 0.42 | 0.66 | 0.75 | 0.58 |
| siglip | 0.54 | 0.81 | 0.87 | 0.64 |
| vggt | 0.47 | 0.51 | 0.56 | 0.49 |
| **Mean (Linear)** | **0.48** | **0.69** | **0.75** | **0.62** |
| **NON-LINEAR PROBE** | | | | |
| clip | 0.53 | 0.69 | 0.77 | 0.68 |
| convnext | 0.59 | 0.69 | 0.80 | 0.71 |
| deit iii | 0.50 | 0.71 | 0.72 | 0.68 |
| dino | 0.63 | 0.67 | 0.70 | 0.66 |
| dinov2 | 0.67 | 0.91 | 0.81 | 0.75 |
| dreamsim | 0.60 | 0.62 | 0.60 | 0.68 |
| dust3r | 0.58 | 0.55 | 0.55 | 0.54 |
| mae | 0.61 | 0.66 | 0.61 | 0.60 |
| mast3r | 0.57 | 0.56 | 0.56 | 0.52 |
| sam | 0.52 | 0.62 | 0.71 | 0.59 |
| siglip | 0.69 | 0.78 | 0.84 | 0.73 |
| vggt | 0.52 | 0.54 | 0.53 | 0.50 |
| **Mean (Non-Linear)** | **0.56** | **0.67** | **0.70** | **0.64** |


**2. Evaluation of 3D-Native VLMs**

To determine if the observed performance gap was strictly due to 2D inputs, we evaluated 3D-native vision-language models using both single-view images (LLaVA-3D) and ground truth point clouds (ShapeLLM, PointBind&PointLLM).
| Method | Platonic | Archim. | Catalan | Johnson | KP | Stel. | Comp. | NonConv. |
| :--- | :---: | :---: | :---: | :---: | :---: | :---: | :---: | :---: |
| **SIMPLE PROMPT** | | | | | | | | |
| LLaVA-3D (Syn) | 20.0% | 0.0% | 0.0% | 0.0% | 0.0% | 0.0% | 0.0% | 0.0% |
| LLaVA-3D (Wild) | 20.0% | 0.0% | 0.0% | 0.0% | 0.0% | 0.0% | 0.0% | 0.0% |
| ShapeLLM | 40.0% | 7.6% | 0.0% | 0.0% | 0.0% | 0.0% | 0.0% | 0.0% |
| PointLLM | 40.0% | 7.6% | 7.6% | 1.0% | 25.0% | 0.0% | 0.0% | 3.8% |
| **CoT PROMPT** | | | | | | | | |
| LLaVA-3D (Syn) | 0.0% | 0.0% | 0.0% | 1.0% | 0.0% | 0.0% | 0.0% | 0.0% |
| LLaVA-3D (Wild) | 20.0% | 7.6% | 0.0% | 0.0% | 0.0% | 0.0% | 0.0% | 0.0% |
| ShapeLLM | 20.0% | 7.6% | 0.0% | 1.8% | 0.0% | 0.0% | 0.0% | 0.0% |
| PointLLM | 0.0% | 15.3% | 0.0% | 0.0% | 0.0% | 0.0% | 0.0% | 1.9% |

**3. Multi-View Inputs and Chain-of-Thought Prompting**

We investigated whether "sub-optimal prompting" or "single-view ambiguity" caused VLM failures. We tested frontier models (GPT-4o, GPT-5-mini) using Multi-View inputs (3 canonical views) and Chain-of-Thought (CoT) prompts explicitly requesting reasoning on convexity and symmetry.

| Model / Condition | Plat | Arch | Cat | John | KP | Stel | Comp | NonConv |
| :--- | :---: | :---: | :---: | :---: | :---: | :---: | :---: | :---: |
| **SYNTHETIC - BASELINE** | | | | | | | | |
| GPT-4o (SV) | 80.0% | 7.6% | 0.0% | 0.0% | 50.0% | 0.0% | 10.0% | 0.0% |
| GPT-4o (MV) | 100.0% | 7.6% | 0.0% | 4.0% | 25.0% | 0.0% | 0.0% | 1.9% |
| GPT-5-mini (SV) | 60.0% | 7.6% | 0.0% | 1.4% | 0.0% | 0.0% | 0.0% | 0.0% |
| GPT-5-mini (MV) | 80.0% | 15.3% | 7.6% | 2.1% | 0.0% | 2.9% | 10.0% | 0.0% |
| **SYNTHETIC - CoT** | | | | | | | | |
| GPT-4o (SV) | 100.0% | 0.0% | 0.0% | 0.0% | 50.0% | 2.9% | 0.0% | 0.0% |
| GPT-4o (MV) | 100.0% | 0.0% | 0.0% | 2.9% | 25.0% | 2.9% | 0.0% | 1.9% |
| GPT-5-mini (SV) | 60.0% | 7.6% | 0.0% | 1.4% | 25.0% | 0.0% | 10.0% | 0.0% |
| GPT-5-mini (MV) | 100.0% | 23.0% | 15.3% | 5.0% | 25.0% | 2.9% | 10.0% | 0.0% |
| **WILD - BASELINE** | | | | | | | | |
| GPT-4o (SV) | 20.0% | 7.6% | 0.0% | 2.7% | 50.0% | 0.0% | 16.7% | 0.0% |
| GPT-4o (MV) | 40.0% | 0.0% | 0.0% | 3.4% | 25.0% | 0.0% | 16.7% | 1.9% |
| GPT-5-mini (SV) | 20.0% | 23.0% | 7.6% | 0.0% | 0.0% | 0.0% | 0.0% | 0.0% |
| GPT-5-mini (MV) | 20.0% | 23.0% | 7.6% | 3.4% | 25.0% | 0.0% | 16.7% | 0.0% |
| **WILD - CoT** | | | | | | | | |
| GPT-4o (SV) | 20.0% | 7.6% | 0.0% | 0.0% | 50.0% | 0.0% | 0.0% | 0.0% |
| GPT-4o (MV) | 40.0% | 15.3% | 0.0% | 2.7% | 25.0% | 0.0% | 16.7% | 1.9% |
| GPT-5-mini (SV) | 0.0% | 0.0% | 7.6% | 2.0% | 25.0% | 0.0% | 0.0% | 0.0% |
| GPT-5-mini (MV) | 0.0% | 30.7% | 15.3% | 6.9% | 50.0% | 0.0% | 16.7% | 0.0% |

---

> ### Author Response · Authors · 2025-11-22
> **Additional experiments (cont.)**
>
> **4. Human Baseline Study (Mental Rotation Test)**
>
> To strictly contextualize model performance on the Mental Rotation Test (MRT), we conducted a controlled user study with 42 participants.
>
> Protocol: We developed a web interface presenting the exact same image pairs used in the paper. Participants answered 25 questions (5 from the "easy" synthetic-synthetic set, 20 from the "hard" synthetic-wild set) with a strict 30-second time limit per question. Options were "Same", "Different", or "I don't know" (scored as 0).
>
> Results:
>
> Easy Set: Mean accuracy was 97.56%, confirming participants understood the task.
>
> Hard Set: Mean accuracy was 68.05% (Std Dev: 0.11), with top performers scoring 90% (18/20).
>
> Comparison: While the best non-linear probe (SigLIP) achieved 69%—matching the average human—68% of individual human participants scored higher than the best performing foundation model. This confirms that while models are closing the gap on average, a significant performance disparity remains on complex instances compared to human capability.
>
> We have uploaded a revised PDF of the main paper and supplementary material incorporating these results with changes highlighted in blue.

---

### Author Response · Authors · 2025-12-01
**Message to AC: New Experiments & Consensus Summary**

Dear Area Chair,

We understand that scores have been reverted due to the recent policy change. We are writing to summarize the substantial new experimental work completed during the rebuttal to address reviewer concerns, and the resulting consensus.

**1. Explicit Reviewer Consensus**
Following our rebuttal, two reviewers explicitly confirmed their concerns were resolved:
* **Reviewer pamt (Initial Score: 8):** Stated, *"I believe the authors addressed my concerns and I am inclined to acceptance."*
* **Reviewer MgEn (Initial Score: 6):** Stated, *"The response addressed the concerns I had in the initial review,"* and explicitly committed to re-evaluating the paper based on our new data: *"I will carefully consider the response, the submitted manuscript, and the fellow reviewers' assessments in making the final decision."*

**2. Extensive New Experiments**
To address critiques regarding 3D baselines and probing methodologies (Reviewer 15Vf), human context (Reviewer pamt), and prompting strategies (Reviewers 15Vf, pamt, GJKD), we conducted four major new studies:

* **A. 3D-Native & Multi-View Evaluation:** We evaluated geometry-native models (LLaVA-3D, ShapeLLM, PointLLM) and multi-view encoders (VGGT, DUSt3R, MASt3R).
    * **Result:** These models also failed significantly (e.g., ShapeLLM <10% accuracy on most classes; VGGT/DUSt3R probes performed worse than 2D baselines).
    * **Implication:** This validates GIQ’s core claim: the failure is a fundamental gap in *geometric reasoning*, not merely a modality limitation of 2D models.

* **B. Non-Linear Probing:** As requested by Reviewer 15Vf, we expanded our analysis to include non-linear probes.
    * **Result:** While non-linear probes improved upon linear baselines, the performance gains were insufficient to bridge the reasoning gap, confirming the issue lies in the representation itself.

* **C. Human Baseline Study (N=42):** We conducted a controlled study with 42 participants on the Mental Rotation task.
    * **Result:** While the best non-linear probe (SigLIP, 69%) matches the *average* human (68%), 68% of individual humans outperformed the best model (with top performers reaching 90%).
    * **Implication:** This establishes a valid "human-level" target that current models have yet to reliably reach.

* **D. Chain-of-Thought & Multi-View Prompting:** Addressing concerns from Reviewers 15Vf, pamt, and GJKD, we tested GPT-4o and GPT-4o-mini with CoT prompts and multi-view inputs.
    * **Result (CoT):** Explicit reasoning prompts did not significantly improve accuracy; models frequently hallucinated intermediate geometric features (e.g., non-existent faces) to justify incorrect classifications.
    * **Result (Multi-View):** Multi-view inputs yielded small improvements on low-symmetry shapes (Johnson solids, 0% -> 4%) However, they yielded no significant gains for high-symmetry groups (Platonic, Archimedean, Compound, Nonconvex, Catalan), confirming that for these shapes, a single canonical view is informationally complete but models still fail to reason about it.

We have uploaded a revised manuscript and supplementary material with these new results and analyses highlighted in blue.

We are happy to provide further details if needed.

---

### Meta-Review · Area_Chair_GhuN · 2025-12-09

**Summary:**

This work introduces a new benchmark for evaluating the geometric reasoning capabilities of vision and vision–language foundation models using synthetic and real-world images paired with 3D meshes of diverse polyhedra. Experiments on monocular 3D reconstruction, symmetry detection, mental rotation, and zero-shot classification reveal that existing 2D and 3D foundation models perform poorly. Earlier concerns about missing evaluations—such as 3D-native and multi-view baselines, non-linear probing, and Chain-of-Thought prompt design—have been largely addressed by newly added experiments. The remaining concern is that the highly regular and stylized nature of the polyhedra may not reflect the complexity of real-world objects, making some results expected given the limited exposure of current models to such shapes. The work would benefit from deeper discussion of the insights gained and their implications for building stronger 3D foundation models.

**Reviewer Concerns:**

Reviewer 15Vf raised concerns regarding the absence of evaluations using 3D-native and multi-view models, non-linear probing, and Chain-of-Thought (CoT) prompts; these issues are largely addressed by the authors’ newly added experiments. Reviewer pamt also noted limitations in prompt design, which the authors have mitigated through additional CoT-based evaluations. Reviewers GJKD and MgEn questioned the real-world applicability of the benchmark, emphasizing that the highly regular and stylized polyhedra used may not reflect the complexity and irregularity of real-world objects. While the authors provide reasonable justification for the benchmark’s relevance, their response still lacks deeper insights into how the findings could guide future research toward building more robust 3D foundation models.

**Reviewer Scores:**

Reviewer 15Vf assigned an initial score of 4 and may increase it to 6, as most concerns appear to be resolved by the newly added experiments. Reviewer pamt gave an initial score of 8, raised only minor issues, and is likely to maintain the score, noting that the authors have adequately addressed all concerns and expressing an inclination toward acceptance. Reviewer GJKD assigned a score of 6 and is likely to keep it unchanged; while concerns about prompt ambiguity have been largely mitigated through additional CoT experiments, the primary issue remains the limited real-world applicability of the benchmark due to the stylized nature of the polyhedra. Reviewer MgEn also gave an initial score of 6 and will likely retain it, similarly emphasizing that the regular, stylized polyhedra limit real-world relevance; although the authors justify the benchmark’s value, their response lacks deeper insights into how the findings could guide the development of more robust 3D foundation models.

---

### Decision · Program_Chairs · 2026-01-26

Accept (Poster)